# Stable GNN Embeddings for Relational Data

## Abstract

Graph neural networks (GNNs) are a valuable tool for extracting meaningful representations from graph-structured data. Graphs, like relational databases, represent relationships between entities. Recent research has explored the potential of using GNNs for downstream tasks on relational data, such as entity resolution and missing value imputation. However, applying GNNs to relational databases presents two challenges. The first challenge is data conversion: relational databases, organized as tables connected by key / foreign key constraints, must be transformed into graphs without losing essential information. The second challenge is ensuring that the embedding technique can adapt to the dynamic nature of databases. When a database is updated, the embeddings of the resulting database should be recomputable efficiently. This requires that previously computed embeddings remain stable despite changes to the data. Motivated by using GNNs for relational databases, we study *stability*, i.e., how much the embeddings generated by a GNN change when the input graph undergoes modifications. Building upon the work of Gama et al. (2020), which established a limit for the distance between embeddings of similar graphs, we focus on *node-level stability* for GNN embeddings, particularly when the graphs originate from relations. We propose several techniques for transforming relational databases into graphs. To assess the effectiveness of these methods, we conduct experiments using the TPC-E database benchmark and analyze their stability.

## 1 Introduction and Motivation

Graphs offer a powerful representation of real-world data, capturing intricate relationships and dependencies. However, many modern databases still adhere to the relational data model, popularized by Codd decades ago; according to one source,[1] a whopping 72% of the world's database management systems are relational as of June 2022, while by contrast, a mere 1.8% are graph DBMS. Given the predominance of enterprise and other operational data in the form of relational databases, many predictive tasks must operate directly on this relational data.

Recent advances in deep learning, particularly Graph Neural Networks (GNNs), have demonstrated their effectiveness in learning from tabular, relational data. Our research focuses on "in-database" learning, where GNNs are applied directly to relational data within operational databases, after transforming the relational data into a graph representation. This approach is motivated by the ubiquity of relational databases and by the dynamic nature of operational data, which raises unique challenges compared to static data like images or text.

Several recent studies have explored graph representations of relational databases. By transforming relational data into graphs, these studies have successfully applied GNNs to address downstream tasks like entity resolution and missing value imputation within the original relational data. However, the existing studies often overlook the *stability* of graph representations of relational data and of the corresponding GNN models. In general, small changes in the underlying data can lead to significant shifts in the GNN-learned representations, potentially resulting in drastically different predictions and thus different outcomes in downstream tasks. Therefore, techniques to understand and mitigate or bound these effects are essential.

Inspired by the successful application of GNNs to relational databases, we investigate GNN *stability*. This involves understanding how changes to the graph representation of a relational database affect

---

[1] https://tinyurl.com/popularDBMS

the generated embeddings. Building upon the work of Gama et al. (2020), we revisit the influence of graph topology on GNN outcomes and establish a boundary for the distance between embeddings of similar graphs. Gama et al. (2020)'s study primarily focuses on small graph perturbations, overlooking the potential impact of more significant changes often seen in relational databases, such as modifying attribute values or adding /removing tuples. To address this issue, we modify the specific type of GNN considered in Gama et al. (2020) and derive a global stability theorem based on their local stability results.

Our **first contribution** is hence to extend the stability results to take into account more significant graph perturbations, and to provide bounds that do not directly depend on the size of the input graph (Section 4); both of these properties are desirable in the context of relational databases.

Our **second contribution** is a comprehensive analysis of existing *heterogeneous* (having different types of nodes) database-to-graph constructions. We identify one major drawback, their tendency to be overly *heterophilic* (with dissimilar nodes grouped together), a characteristic that can hinder the effectiveness of GNNs Zhu et al. (2021),He et al. (2022), Wang et al. (2022b). To address it, we propose an alternative construction method (detailed in Section 5). Our experiments on the Cora and TPC-E datasets demonstrate that our approach is up to 10 times more stable than existing state-of-the-art methods (Section 6).

## 2 RELATED WORK

**Graph Neural Networks**  GNNs are a robust and versatile tool for prediction tasks related to graph-structured data. Inspired by Convolutional Neural Networks (CNNs), GNNs compute information for each node by aggregating data from neighboring nodes. Many GNN architectures have been proposed; these include GNNs with attention mechanisms Veličković et al. (2017), simple convolution Kipf & Welling (2016), spectral convolution Defferrard et al. (2016), inductive learning Hamilton et al. (2017), or message passing approaches Gilmer et al. (2020), Battaglia et al. (2016).

**Stability of GNNs**  When working with GNNs, it is essential to consider how their output is influenced by the structure of the input graph. A desirable property is *isomorphism invariance*: when two graphs are identical in structure (isomorphic), a GNN should produce the same embeddings.

The concepts of *invariant* and *equivariant* GNNs have been investigated by Maron et al. (2018), focusing on ensuring that the network's output remains unchanged when nodes are permuted. Keriven & Peyré (2019) proposed universal GNNs, which achieve permutation invariance by using message-passing frameworks, guaranteeing that the network's behavior is not affected by the order of nodes.

While permutation equivariance is a valuable property for GNNs, it does not fully explain how embeddings change when nodes or edges are added or removed. Diffusion-based GNNs, introduced by Gama et al. (2018), focus on ensuring stability under local graph perturbations in diffusion processes. Subsequent work by Gama et al. (2020) derived stability guarantees for GNNs, focusing on how small perturbations in graph structure affect GNN performance, ensuring robust feature extraction from locally perturbed graphs. The work of Wang et al. (2022a) has further explored GNN stability, including the use of positional encoding. Zou & Lerman (2020) introduced graph wavelet filter banks to achieve robustness against structural changes; they ensure stability by filtering graph signals and capturing multi-scale information without the need for supervised learning. Finally, Böker et al. (2024) extended the Weisfeiler-Lehman test to graphons, giving a more general framework for analyzing stability in large graphs. Another point of view of stability, when the graphs are unchanged, but instead the features applied to the GNN vary, is studied in Jia & Zhang (2023).

Another line of research focuses on GNN robustness against adversarial attacks. Such attacks involve making subtle changes to the input data with the intention of misleading the GNN into making incorrect predictions or classifications. To defend against such attacks, Zügner et al. (2018) proposed robust training techniques that enhance the resilience of GNNs. Ennadir et al. (2024) explored the use of noise injection to mitigate the impact of adversarial perturbations. Nikolentzos et al. (2024) discussed the vulnerability of GNNs to attacks targeting both the graph structure and node attributes, highlighting the need for robust defenses.

**Generating embeddings for relational databases**  Several approaches have been developed to create embeddings for relational databases using language models. Fey et al. (2023) introduced a

modular neural message-passing system that can operate directly on relational databases, aligning with the formal relational model and enabling deep learning directly in database systems, without the need for costly data pre-processing stages.

Other methods focus on constructing heterogeneous or n-partite graphs to generate relevant embeddings. Zahradník et al. (2023) proposed Relational Deep Learning (RDL), which treats relational databases as heterogeneous graphs and employs Message Passing Graph Neural Networks (MPNNs) to learn from interconnected data. Similarly, Cappuzzo et al. (2024) proposes an approach for missing value imputation using GNNs on heterogeneous graphs, based on techniques designed for heterogenous graphs such as those in Schlichtkrull et al. (2018).

Some methods utilize homogeneous graphs. Toenshoff et al. (2023) generated a bipartite graph and employed random walk methods to create embeddings, verifying their stability by the accuracy of downstream tasks. Cappuzzo et al. (2020) proposed a tripartite graph representation of relational databases, while Niepert (2016) used a Gaifman graph for generating embeddings.

Many existing homogeneous graph representations of databases exhibit a heterophilic property, where nodes of different types are more likely to be connected, which can hinder the generation of meaningful embeddings Abu-El-Haija et al. (2019), Zhu et al. (2020), Zhu et al. (2021), Chien et al. (2020). We present in this paper graph constructions that alleviate this problem.

## 3 GRAPH NEURAL NETWORKS

### 3.1 PRELIMINARIES

**Graphs** In this section we consider graphs, $\mathcal{G} = (V, E)$ consisting of a finite set $V$ of $N$ nodes and set of edges $E \subseteq V \times V$. On this graph, we can define *graph shift operators* (GSO), matrices $\boldsymbol{S} \in \mathbb{R}^{N \times N}$ with the property that $\boldsymbol{S}_{ij} = 0$ if $(i, j) \notin E$ and $i \neq j$. The adjacency matrix $\boldsymbol{A}$ and the Laplacian matrix $\boldsymbol{L}$ are two examples of GSOs.

**MPNNs** Combination-Aggregation GNNs, a.k.a. Message Passing Neural Networks (MPNN) Gilmer et al. (2017), operate by iteratively aggregating information from a node's neighbors and updating the node's representation through a combination of this aggregated information and the previous node information. The two main steps are message passing and update, represented as:

$$\eta_v^{(l)} = \text{UPDATE}^{(l)} \left( \eta_v^{(l-1)}, \sum_{u \in \mathcal{N}(v)} \text{MESSAGE}^{(l)} \left( \eta_v^{(k-1)}, \eta_u^{(l-1)}, e_{uv} \right) \right), \tag{1}$$

where $\eta_v^{(l)}$ is the representation of node $v$ at layer $l$, $\mathcal{N}(v)$ is the set of neighbors of $v$, and $e_{uv}$ is the edge feature between nodes $u$ and $v$. MESSAGE$^{(l)}$ and UPDATE$^{(l)}$ are learnable functions.

From this general formulation of GNNs, different variants were proposed. In GATs Veličković et al. (2017) attention mechanisms are applied to learn the importance of neighboring nodes. GCNs Kipf & Welling (2016) simplify the message-passing framework by using normalized adjacency matrices for aggregation. The key aim is to smooth the node features based on their neighbors. Another variant are GINs Xu et al. (2018) designed to output embeddings as powerful as the Weisfeiler-Lehman graph isomorphism test, and where the aggregation function uses a multi-layer perceptron (MLP) to update node embeddings.

### 3.2 SIGNAL PROCESSING GNNS (SPGNNS)

We detail the variant of GNNs that we study, namely Signal Processing GNNs (SPGNNs), inspired by signal processing techniques. The following definitions are adapted from Gama et al. (2020).

**Features** With each layer $l$, we associate $F^{(l)}$ vectors, called features or *graph signals*, denoted $(\boldsymbol{x}_f^{(l)})_{f \in \{1, \dots F_l\}} \in (\mathbb{R}^N)^{F_l}$ and $\boldsymbol{X}^{(l)} \in \mathbb{R}^{N \times F_l}$ the feature map composed by all the $F^{(l)}$-th graph signals. The features are computed from the ones from the previous layer $l - 1$.

**Construction** To compute $\boldsymbol{X}^{(l+1)}$ from $\boldsymbol{X}^{(l)}$, we define a sequence of matrices $(\boldsymbol{H}^{(l,k)})_{k\in\mathbb{N}} \in \mathbb{R}^{F^{(l)}\times F^{(l+1)}}$ that is finite, i.e. $\boldsymbol{H}^{(l,k)} = 0$ for $k > K$. The matrices contain the learnable parameters $(\boldsymbol{H}^{(l,k)}_{g,f})_{(g,f)\in F^{(l)}\times F^{(l+1)}}$ of the network. From these parameters, a family $(\boldsymbol{z}^{(l+1)}_f)_{f\in\times\{1,...,F^{(l+1)}\}}$ of intermediate vectors is then computed: $\boldsymbol{z}^{(l+1)}_f = \sum_{k=0}^{K}\sum_{g=1}^{F^{(l)}} \boldsymbol{S}^k \boldsymbol{x}^{(l)}_g \boldsymbol{H}^{(l,k)}_{g,f}$ where $\boldsymbol{x}^{(l)}_g$ is $\boldsymbol{X}^{(l)}_{:,g}$. If we denote by $H^{(l)}_{f,g}$ the polynomial $\sum_{k=0}^{K} \boldsymbol{H}^{(l,k)}_{f,g} X^k$, the formula can be written $\boldsymbol{z}^{(l+1)}_f = \sum_{g=1}^{F^{(l)}} H^{(l)}_{g,f}(\boldsymbol{S})\boldsymbol{x}^{(l)}_g$. The polynomials $H^{(l)}_{f,g}$ are called *filters*. The matrix $\boldsymbol{X}^{(l+1)}$ is then defined by its columns $\boldsymbol{X}^{(l+1)}_{:,f} = \sigma(\boldsymbol{z}^{(l+1)}_f)$ where $\sigma$ is a pointwise non-linearity (e.g., ReLU). This can be written in a matrix form:

$$\boldsymbol{X}^{(l+1)} = \sigma(\sum_{k=0}^{K}\boldsymbol{S}^k\boldsymbol{X}^{(l)}\boldsymbol{H}^{(l,k)}).$$

We denote by an $L$-layer SPGNN $\Phi$ a function composed by the $L$ layers defined above, taking as input a Graph Shift operator (GSO) $\boldsymbol{S}$ and a graph signal $\boldsymbol{x} \in \mathbb{R}^N$. The $\boldsymbol{X}^{(0)}$ matrix is initialised as the single-column matrix containing the vector $\boldsymbol{x}$, while the output is the only vector stored in the last layer, $\boldsymbol{X}^{(L)}_{:,1}$. In particular, $F_0 = F_L = 1$.

**Node embeddings** Strictly speaking, the $L$-layer SPGNN $\Phi$ as defined above does not output embeddings: when applied to a vector, it returns one *coordinate* of the final embedding for each node. To return a $d$-dimensional representation of each node, we use the SPGNN on $d$ vectors in $\mathbb{R}^N$ and concatenate the results, which is equivalent to applying the SPGNN on a $N \times d$ matrix.

We henceforth denote as $\boldsymbol{X} \in \mathbb{R}^{N\times d}$ the initial feature matrix of the graph, where each row corresponds to the initial embedding of a node, and $\boldsymbol{X}' \in \mathbb{R}^{N\times d}$ as the final feature map of the graph, where its $i$-th column ($i \leq d$) is the $i$-th column of $X$ after transformation via $\Phi$.

**Computational issues** Computing the final feature map as shown previously requires recomputing the $H$ matrices of all the graph filters. This poses a significant memory problem if we want to carry out learning with high-dimensional embeddings, as all the matrices computed in this way will have to be saved during the process. This is why we have restricted the SPGNN method to low dimensional representations. Despite this apparent shortcoming, our experiments, in Section 6, show that SPGNN remains competitive compared to the state-of-the-art GNNs.

## 4 STABILITY OF SPGNNS

In this section, we discuss the main technical results of this paper. We first present a refined bound on the stability of SPGNNs, then we discuss the stability of node embeddings in the average case, and finally we provide a refined bound for the stability of node embeddings in the worst case.

### 4.1 EXISTING RESULTS

We use the norm operator of a matrix $\boldsymbol{S}$, denoted $||\boldsymbol{S}||_{op}$ and defined as $||\boldsymbol{S}||_{op} := \max_{||x||=1} ||\boldsymbol{S}x||$.

The final theorem in Gama et al. (2020), particularly the first point of their Property 3, can be simplified as follows:

**Theorem 1** (Theorem 4 & Property 3 in Gama et al. (2020)). Let $\boldsymbol{S}$ and $\hat{\boldsymbol{S}}$ be two GSOs and $\Phi$ be a SPGNN. We assume that, for all filters $H$ appearing in $\Phi$ and all eigenvalues $\lambda$ of $\boldsymbol{S}$ and $\hat{\boldsymbol{S}}$, $|H(\lambda)| \leq 1$. We also assume that the pointwise non-linearities in $\Phi$ are 1-Lipschitz. Define $\varepsilon := ||\boldsymbol{S} - \hat{\boldsymbol{S}}||_{op}$ and $\delta = (||U - V||^2 + 1)^2 - 1$ where $U$ and $V$ are orthogonal matrices s.t. $U^T S U$ and $V^T(S - \hat{S})V$ are diagonal. Then, the following inequality holds for all $\boldsymbol{x} \in \mathbb{R}^N$ s.t. $||\boldsymbol{x}|| = 1$:

$$||\Phi(\boldsymbol{S}, \boldsymbol{x}) - \Phi(\hat{\boldsymbol{S}}, \boldsymbol{x})|| \leq (1 + \delta\sqrt{N})LF^{L-1}\varepsilon + \mathcal{O}(\varepsilon^2).$$

While this result is interesting and has been a model for our theoretical work, we face several limitations when trying to apply it. Firstly, the result is local, i.e., only relevant when the amplitude of the perturbation $\varepsilon = ||S - \hat{S}||$ goes to 0. Yet the perturbations we are interested in – in the context of relational databases – can produce modification of graphs that *cannot* be considered as infinitesimal. For example, a tuple being removed would result in the deletion of several edges in the graph representation of the database; the perturbation, measured as $||\boldsymbol{S} - \hat{\boldsymbol{S}}||_{op}$, would typically be roughly equal to the number of deleted edges. Moreover, the assumption on the filters ($|H(\lambda)| \leq 1$) may not always hold, since the parameters of the filters are the result of the SPGNN training and therefore are hard to control. Finally, a key limitation of Gama et al. (2020) is the dependence on the graph size ($N$), making it difficult to establish results on asymptotic stability, particularly when dealing with large graphs and ensuring that embeddings remain reasonably stable as the data grows.

In the next section, we establish a result (Theorem 2) that addresses these issues by using a degree 1 assumption on the polynomial defining the filters. We also show that this assumption can generalize.

### 4.2 REFINED BOUNDS ON THE STABILITY OF SPGNNs

Our main theorem provides a refinement of the upper bound in Gama et al. (2020) on the distance between the embeddings of two graphs. The main improvement is that our bound is global and does not depend on $N$, the size of the graph, although the bound involves $N$-dimensional norms:

**Theorem 2.** Let $\Phi(\cdot, \cdot)$ be a GNN with $L$ layers, with fewer than $F$ features per layer, that only uses filters of the form $H(X) = aX + b$. We assume that the non-linearities $\sigma$ in $\Phi$ are 1-Lipschitz and satisfy $\sigma(0) = 0$. Let $\boldsymbol{S}, \hat{\boldsymbol{S}} \in \mathbb{R}^{N \times N}$ be two symmetric matrices. Let $||H||_{\infty} := \max\limits_{H \in \Phi, \ \lambda \in \sigma(\boldsymbol{S}) \cup \sigma(\hat{\boldsymbol{S}})} |H(\lambda)|$, the maximum of the value $|H(\lambda)|$ over all filters $H$ that appear in $\Phi$ and all eigenvalues $\lambda$ of $\boldsymbol{S}$ or $\hat{\boldsymbol{S}}$. Let $A := \max\limits_{H(X) = aX + b \in \Phi} |a|$.

Then, for all $\boldsymbol{x} \in \mathbb{R}^N$, we have:

$$||\Phi(\boldsymbol{S}, \boldsymbol{x}) - \Phi(\hat{\boldsymbol{S}}, \boldsymbol{x})|| \leq LA(F||H||_{\infty})^{L-1}||\boldsymbol{S} - \hat{\boldsymbol{S}}||_{op}||\boldsymbol{x}||. \tag{2}$$

The proof can be found in Appendix A.1.

**Remark 1** The bound given in Theorem 1 is useful to understand how the stability evolves depending on the parameters of the SPGNN. However, our proof reveals an even tighter bound, albeit less readable and less intuitive.

**Remark 2** Note that, in Eq. 2, the value of $||H||_{\infty}$ depends on $\boldsymbol{S}$ and $\hat{\boldsymbol{S}}$. Therefore, this equation does *not* show that SPGNNs are Lipschitz. However, if we restrict the domain of $\boldsymbol{S}, \hat{\boldsymbol{S}}$ to a bounded subset of $\mathbb{R}^{N \times N}$, then Eq. 2 indeed states that SPGNNs are Lipschitz. For instance, if $\boldsymbol{S}, \hat{\boldsymbol{S}}$ are adjacency matrices, then their Frobenius norm (defined as $||\boldsymbol{S}||_F := \sqrt{\text{tr}(A^T A)}$) is less than $N$, so their eigenvalues are less than $N$. Therefore, $||H||_{\infty}$ can be bounded by $\max\limits_{\lambda \in [-N,N], H_{i,j}^{(l)} \in \Phi} |H_{i,j}^{(l)}(\lambda)|$ and then Theorem 2 does imply that, for a given $x \in \mathbb{R}^N$, $\Phi(\cdot, x)$ is a Lipschitz function.

**Remark 3** The result in Theorem 2 can be instantiated in the *distance modulo permutation* framework used in Gama et al. (2020). For the sake of clarity and due to lack of space, we defer this to the supplementary material, Appendix A.3.

**Remark 4** We chose to state Theorem 2 for SPGNNs that only contain filters that are order 1 polynomials because a SPGNN that uses polynomials of higher order can be reduced to a SPGNN that only uses filters of order 1. This is detailed in Appendix A.4.1 and illustrated in Figure 4.

However, to apply our results directly to a SPGNN $\Psi$ with $L$ layers, fewer than $F$ features by layer, and using higher order polynomials, the following holds, in the same vein as Theorem 2:

$$||\Psi(\boldsymbol{S}, \boldsymbol{x}) - \Psi(\hat{\boldsymbol{S}}, \boldsymbol{x})|| \leq L\Delta(H)_{\infty}(F||H||_{\infty})^{L-1}||S - \hat{S}||_{op}||\boldsymbol{x}||$$

where $\Delta(H)_\infty = \max\limits_{H \in \Psi} \Delta(H)$ and if $H(X) = \sum\limits_{i=0}^{d} h_i X^i$, $\Delta(H) := \sum\limits_{i=1}^{d} i|h_i|\Lambda^{i-1}$ for $\Lambda = \max\limits_{\lambda \in \sigma(S) \cup \sigma(\hat{S})} |\lambda|$. The proof is given in Appendix A.4.1.

As a consequence, our framework can be applied to a wide variety of architectures, such as graph convolutional neural networks (see also Remark 3 in Gama et al. (2020)).

**Remark 5** While this is not our focus in this work, we mention that Theorem 2 extends to non-symmetric matrices, i.e. to directed graphs. This is further detailed in Appendix A.4.2.

### 4.3 NODE EMBEDDING STABILITY

The result of Theorem 2 holds for the embeddings of all the nodes of the graph, but it may be too loose for some node pairs. For instance, for a graph $\mathcal{G}$ and $\hat{\mathcal{G}}$ obtained from $\mathcal{G}$ by removing an edge, the bound on the distance between two node embeddings is the same for a node that "lost" that edge and for a node that is far from the deleted edge. However, intuitively we would expect that for nodes that are far from the perturbation the effect on their embeddings should be less pronounced.

Therefore, in this section, we discuss two corollaries of Theorem 2 that translate the original result in stability properties at the node embedding level and deal with the refinement of the bound for nodes that are less impacted by the perturbation.

Notice that being given a graph shift operator (GSO) on a graph amounts to being given integers corresponding to an arbitrary but fixed ordering of the nodes: each node $v$ of $\mathcal{G}$ is represented by an integer $i$ that corresponds to the row/column of $S$ that represents this node. If we are given two graphs $\mathcal{G}, \hat{\mathcal{G}}$ with two associated GSOs $S, \hat{S}$, then we can define the identification of two nodes by stating that a pair of nodes in $(\mathcal{G}, \hat{\mathcal{G}})$ are *identified* if they are represented by the same integer $i$. When discussing the node embedding stability, we always compare the distance between the embeddings of a pair of identified nodes, and, in the following, we use the notation $(v_i, \hat{v}_i)$ to refer to the couple of nodes in $\mathcal{G} \times \hat{\mathcal{G}}$ that are identified by the integer $i$.

The purpose of the first corollary is to instantiate Theorem 2 to bound the distance between node embeddings. The bound is given on the sum of these distances, which provides both a worst case bound and an average bound:

**Corollary 2.1.** Consider a GNN $\Phi$ satisfying the hypotheses of Theorem 2, and a pair of graphs $\mathcal{G}, \hat{\mathcal{G}}$ with their associated shift operators $S, \hat{S}$. Assume that all the columns of the initialisation matrix $X^{(0)}$ are normalized. Then the total squared distances between the embeddings satisfies:

$$\sum_{i=1}^{N} ||\Phi(S, X^{(0)})_{v_i} - \Phi(\hat{S}, X^{(0)})_{\hat{v}_i}||^2 \le dM^2, \tag{3}$$

where $M$ is the bound of Theorem 2, i.e. $M = B^{(L)}||S - \hat{S}||_{op}$.

The proof can be found in appendix A.2.

**Remarks** On the one hand, Corollary 2.1 implies that, given a pair of identified nodes $(v_i, \hat{v}_i)$, we can bound the distance of their embeddings by $||\Phi(S, X^{(0)})_{v_i} - \Phi(\hat{S}, X^{(0)})_{\hat{v}_i}|| \le \sqrt{d}M$. On the other hand, the result ensures that, for at least $N/2$ pairs of nodes $(v_i, \hat{v}_i)$, $||\Phi(S, X^{(0)})_{v_i} - \Phi(\hat{S}, X^{(0)})_{\hat{v}_i}|| \le \frac{\sqrt{2d}M}{\sqrt{N}}$. This last inequality ensures that, if we consider GSOs whose eigenvalues tend not to increase with the number of nodes $N$, the bound on the distance between the embeddings approaches 0, which ensures an asymptotic stability for SPGNNs. The fact that the bound no longer depends on $\sqrt{N}$, following our improvement, is crucial for this result to hold.

While Corollary 2.1 states that the majority of the node embeddings are stable, it does not give a method to identify which pairs of nodes will be more stable. The goal of the next result is to analyze pairs of nodes that are more stable than what is stated by Theorem 2. We start by defining the notion of $k$-hop equality, which is crucial to identify these pairs:

**Definition 4.1** ($k$-hop equality). We consider two graphs $\mathcal{G}, \hat{\mathcal{G}}$ with two associated GSOs $\boldsymbol{S}, \hat{\boldsymbol{S}}$. We define the $k$-hop equality relative to $\boldsymbol{S}, \hat{\boldsymbol{S}}$ inductively:

- Every pair of identified nodes in $\mathcal{G}, \hat{\mathcal{G}}$ have equal 0-hops.

- Let $k \geq 1$. If $u$ and $\hat{u}$ are a pair of identified nodes both represented by the same integer $i$, they have equal $k$-hop if $S_{i,:} = \hat{S}_{i,:}$ and all their neighbors can be identified pairwise with each pair having equal $(k-1)$-hops.

In other words, the fact that two nodes $u, \hat{u}$ have equal $k$-hop means that the sub-graph induced in $\mathcal{G}$ by all the edges and vertices that can be reached in a $k$-walk starting from $u$ is exactly the same as the sub-graph induced in $\hat{\mathcal{G}}$ by all the edges and vertices that can be reached in a $k$-walk starting from $\hat{u}$, i.e., the perturbation in $\mathcal{G}$ is at least at distance $k$ from $u$.

Building upon this notion, we can refine the worst-case result of Corollary 2.1 for pairs of nodes that have equal $k$-hops:

**Corollary 2.2.** Consider an SPGNN $\Phi$ satisfying the hypotheses of Theorem 2, and a pair of graphs $\mathcal{G}, \hat{\mathcal{G}}$ with associated shift operators $\boldsymbol{S}, \hat{\boldsymbol{S}}$. Assume that all the columns of the initialisation matrices $\boldsymbol{X}^{(0)}$ are normalized. Consider the $i$-th nodes of $\mathcal{G}$ and $\hat{\mathcal{G}}$, denoted $v_i$ and $v_i'$ and assume that their have equal $k$-hops, then we have the following property:

$$||\Phi(\boldsymbol{S}, \boldsymbol{X}^{(0)})_{v_i} - \Phi(\hat{\boldsymbol{S}}, \boldsymbol{X}^{(0)})_{v_i'}|| \leq \sqrt{d}(L-k)^+ A(F||H||_\infty)^{L-1}||\boldsymbol{S} - \hat{\boldsymbol{S}}||_{op} \qquad (4)$$

where $(L-k)^+ = \max(0, L-k)$.

The proof can be found in Appendix A.2.

In particular, note that when $k \geq L$ we have $\Phi(S, X^{(0)})_{v_i} = \Phi(\hat{S}, X^{(0)})_{v_i'}$. Similar to the findings in Theorem 2, the proof reveals a tighter bound. While this refined bound provides greater accuracy, it may be less intuitive to understand.

## 5 APPLICATION: STABILITY OF EMBEDDINGS FOR RELATIONAL DATABASES

One application scenario where stability is important are databases. This is because databases have frequent updates and even may contain missing values. To ensure that embeddings results from graph constructions from databases can be used even under perturbation, they must be stable.

**Transforming Relational Databases to Graphs** Consider a database $\mathcal{D} = (\Delta, \mathcal{R})$ with a set of tuples $\Delta$, a set of relations $\mathcal{R}$, and the set of its attribute names $\Omega$. The active domain of the database is denoted $adom(\mathcal{D})$, to which we add the NULL value, denoted $\perp$.

There are multiple ways to encode relational databases as graphs. One approach from the state-of-the-art is **EmBDi** Cappuzzo et al. (2020), which constructs a tripartite graph having three types of nodes corresponding to tuples in $\Delta$, values in the database, and attributes in $\Omega$ respectively. Edges exist between values and tuples, and between values and attributes As this graph has three types of node and is tripartite, EmBDi constructs a heterogeneous and heterophilic graph. Another recent construction is **FoRWaRD** Toenshoff et al. (2023), a bipartite graph linking attribute value nodes to the corresponding tuple nodes of the same relation. A value node from a relation is be linked to a tuple node of another relation only if there are foreign key constraints between the two relations. For multiple relations, an edge is present if there are foreign key constraints between them. FoRWard is a heterogeneous and heterophilic graph, containing two types of nodes (hence a bipartite graph). A limitation of these two graph constructions is their heterophilic nature and their inability to fully capture the underlying structure of the database.

To address these limitations, we propose **5-DB**, a novel graph encoding approach for relational databases, designed to more comprehensively capture the underlying database structure.

**Definition 5.1** (**5-DB** construction). Let $\mathcal{D} = (\Delta, \mathcal{R})$ be a database, and $\Omega$ its set of attributes. We create a graph as follows.

*(Nodes).* For each relation $R \in \mathcal{R}$, we create a node $v_R$ with label $R$. For each non-null element $a \in adom(\mathcal{D}) \setminus \{\perp\}$, we create a node $v_a$ with label $a$. For each tuple $t$ in a relation $R \in \mathbb{R}$, we

create a node $v_t$ with label $t$. For each attribute name $A \in \Omega$, we create a node $v_A$ with label $A$. For each relation $R \in \mathbb{R}$, each attribute $A \in \Omega(R)$, and each tuple $t \in R$, we create a node $v_{A,t}$ with label $(A, t)$.

*(Edges)*. For each relation $R \in \mathbb{R}$ and each tuple $t \in R$, we create an edge $(v_R, v_t)$. For each tuple $t$ in a relation $R \in \mathbb{R}$ and attribute name $A$ of the relation $R$, we create $(v_t, v_{A,t})$. For each relation $R \in \mathbb{R}$, each tuple $t \in R$, and each attribute $A \in \Omega(R)$, we create $(v_{A,t}, v_A)$. For each non-null value $a$ of attribute $A$ in tuple $t$, we create $(v_{A,t}, v_a)$. If the value is $\perp$, no edge is created.

**5-DB** has five types of nodes which are never connected to other nodes of the same type. This graph is hence both heterophilic and heterogeneous. We also distinguish a variant of **5-DB**, one in which attribute nodes are removed, denoted **5-DB-Att**. Additionally, we define another variant, **5-DB Gaif**, which is similar to **5-DB-Att** but with an added constraint: values belonging to the same tuple form a clique. **5-DB Gaif**'s rationale is to increase the homophily of the graph. Finally, **Tuple+Clique** is similar to **FoRWaRD**, where the tuples of the same relation form a clique. The goal of this construction is to investigate whether imposing density can enhance the stability of the generated embeddings. All the above graphs are heterogeneous.

**Perturbations**   Our primary motivation for studying GNN embeddings is their application to relational databases, which are often subject to frequent updates (*perturbations*). Two common types of perturbations in databases are:

- *Tuple Removal*: Some tuples present in the original database may be deleted in the updated version. In the corresponding graph representation, this translates to removing nodes associated with these tuples.

- *Value Removal*: Tuples in the updated database might have missing values. This can be represented in the graph by removing edges between attribute nodes and value nodes.

- *Tuple/Value Addition* Since adding tuples/values to a database $\mathcal{D}$ to obtain a database $\hat{\mathcal{D}}$ is the exact opposite of removing tuples/values in $\hat{\mathcal{D}}$ to obtain $\mathcal{D}$, and given that our bounds are symmetric in the matrices $S$ and $\hat{S}$ derived from the databases, the case of tuple/value addition is exactly the same as the case of tuple/value removal. Therefore, in the following, we only consider the case of removal.

From the perspective of this study, adding tuples is essentially the opposite of removing tuples, and adding values is the reverse of removing values. Therefore, to analyze the impact of these operations, we can simply switch the corresponding Graph Shift Operators (GSOs) in our stability formulas.

When a database is modified, its size changes. If we consider directly the corresponding graphs, their GSOs have different sizes, making them incomparable. To circumvent the size mismatch problem, in the case of deletion, we do not truly delete the nodes targeted by the deletion, but we remove all their edges instead. This yields a graph $\hat{\mathcal{G}}'$ with exactly the same node set as $\mathcal{G}$, but with fewer edges. We can therefore directly compute $||S - \hat{S}'||_{op}$. In the case of a tuple addition, we add a row / column of $0$s in the matrix $S$ in the place of the row / column that represents the tuple in the matrix $\hat{S}$. This is one illustration the removal / addition symmetry mentioned before.

Putting it all together, by transforming a relational database into a graph using one of the methods discussed earlier and applying the dimension matching procedure above, we can directly leverage the results of Corollary 2.1 to analyze the stability of tuple embeddings within a database.

## 6   EXPERIMENTAL RESULTS

For our experimental evaluation, we implemented in Python the GNNs that we evaluated along with the graph construction, using the `NetworkX` and `PyTorch Geometric` packages. The experiments were run on a 32 GB machine equipped with a 8GB NVIDIA A2000 GPU. The code and additional implementation and experimental details are available at `https://github.com/ForAnonymousSubmission/STABLE-GNN-EMBEDDINGS-FOR-RELATIONAL-DATA.git`. For space limitations, we present here only the most relevant results, and additional results can be found in Appendix B.

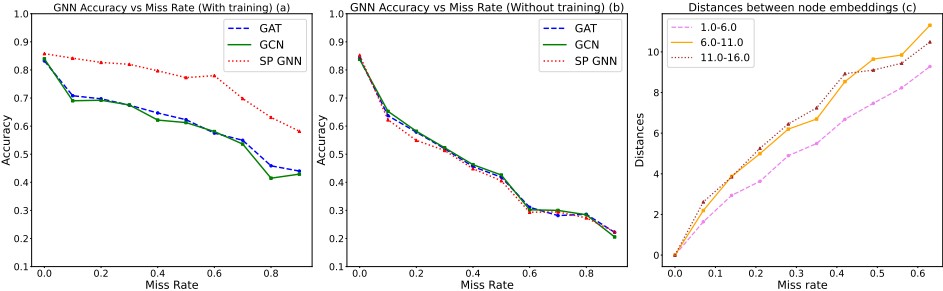

Figure 1: Subfigures (a) and (b): Classification accuracy for the SPGNN having order-one filters and other state-of-the-art GNNs, on the Cora dataset, when removing a proportion of nodes ranging from 0% to 90%. Subfigure (c): Distance between SPGNN embedding for different node degrees.

Table 1: Size of graphs encoding the relational database.

|  | FoRWaRD | EmBDi | 5-DB | 5-DB Gaif | 5-DB-Att | Tuple+Clique |
|---|---|---|---|---|---|---|
| nodes | 10,089 | 10,225 | 26,377 | 26,241 | 26,241 | 10,089 |
| edges | 15,131 | 24,571 | 50,748 | 96,246 | 34,305 | 205,168 |

## 6.1 COMPARING SPGNNS WITH OTHER GNN VARIANTS (CORA GRAPH DATASET)

First, to demonstrate the effectiveness of order-one SPGNNs, we compare them with other GNN architectures. Our results indicate that despite their apparent simplicity, order-one SPGNNs can achieve performance comparable to more complex GNN models. For simplicity and computational efficiency, we use the order-one filter $H(S) = H_1 S + H_0$. We compare SPGNNs with GCNs Kipf & Welling (2016) and GATs Veličković et al. (2017) on the Cora dataset Yang et al. (2016), a graph having 2,708 nodes, 10,556 edges, and 7 node classes. We use the same architecture for all models: a 4-layer GNN followed by a 2-layer MLP for node classification. Each layer of the GNN contains 10-dimensional features. We remove a proportion of nodes from the graph, ranging from 0 to 90% and we report the classification accuracy of the models on the reduced graphs. All GNNs are trained once and then remain fixed; this is because our main objective is to evaluate the stability of the embeddings with respect to perturbations.

The results are shown in Figure 1. Fig. 1(a) shows the accuracy when the classification MLP is re-trained on each incomplete graph. Interestingly, it shows that the accuracy of the SPGNN is more stable than the one of the GCN and GAT. This suggests that the SPGNN embeddings are more robust to perturbations. On the other hand, when the MLP is not re-trained, the three variants are indistinguishable in terms of accuracy, as shown in Fig. 1(b). This suggests that the SPGNN embeddings are not necessarily more informative, in a geometric sense, than those generated by GCN and GAT. The results in Figure 1(c) indicate that the stability of SPGNN embeddings is influenced to some extent by the degree of nodes in the graph. However, this influence is relatively minor, suggesting that SPGNNs can be applied effectively to graphs with varying node degree distributions.

## 6.2 STABILITY FOR GRAPHS ENCODING RELATIONAL DATABASES (TPC-E DATASET)

Having established the stability of SPGNN embeddings, we now evaluate the stability of the different graph constructions introduced in Section 5. To achieve this, we leverage a subset of tables from the well-known TPC-E database benchmark (https://www.tpc.org/) and we construct corresponding graphs for stability analysis. The details of the resulting graphs are given in Table 1, while the statistics on the TPC-E data we used are in Appendix B.

For generating embeddings, we used an 5-layer SPGNN with a polynomial filter of degree one, having the following feature dimensions $F = [3, 4, 6, 3, 1]$, with the last layer of dimension 1 in order to return a graph filter. We initialize a random matrix $X_0 = \mathbb{R}^{N \times d}$ as the initial feature map. The generated embeddings represent different elements from the original database, including tuples, attributes, and values. Our experimental comparison focuses on tuple embeddings since they are present in all graph constructions. Furthermore, stability of tuple embeddings is of fundamental importance since tuples are the core information content in a relational database.

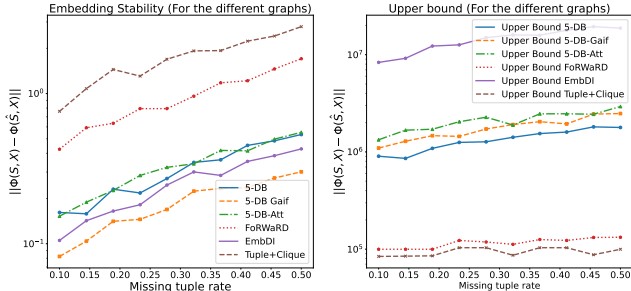

Figure 2: Stability against tuple removal. Left: the average distance between embeddings of the tuples in the original database vs. the perturbed one. Right: the theoretical stability upper bounds.

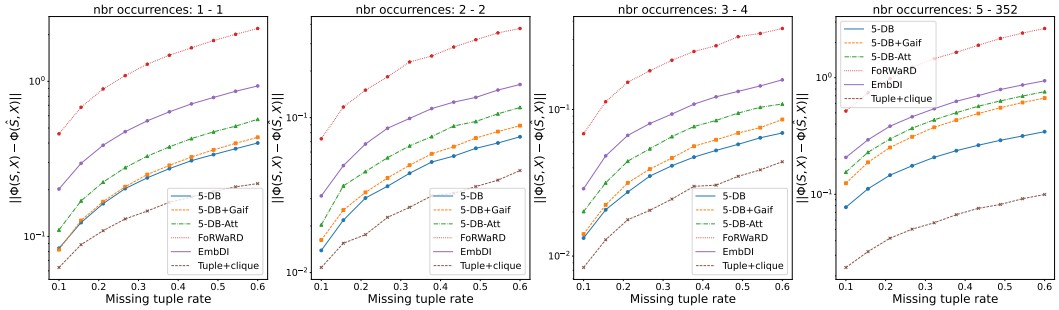

Figure 3: Stability against missing values, by removing up to $90\%$ of the values from the database. We show results for removal of values for various frequency bins.

Figure 2 shows the stability of the embeddings when tuples are removed from the database. We can observe that the SPGNNs used on all the graphs are able to generate stable embeddings. However, graphs that encode explicitly the various relations, using a relation node connected to its tuples (**5-DB**, **5-DB Gaif**, **5-DB-Att**) have lower absolute embedding distances than the other constructions. This suggests that the structure of the graph is important for the stability of the embeddings.

Two relevant results can be observed in Figure 2. First, embeddings generated using **FoRWaRD** and **Tuple+Clique** showed poor resilience to tuple removal. This can be explained by the absence of *anchor* nodes which are less impacted by tuple deletion (for example attribute nodes or relation nodes). Moreover among the graphs generating embeddings showing better resilience to tuple removal, **5-DB Gaif** showed the best results which is not surprising. Indeed, **5-DB Gaif** has an higher density in its subgraphs corresponding to attribute value nodes, which are less likely to be deleted by the process than tuple nodes.

When analyzing the impact of value removal (Figure 3), we observed a similar trend to that of tuple removal. However, **Tuple+Clique** demonstrated more resilience to value removal. This can be attributed to the cliques introduced between tuples of the same relation, resulting in a denser graph structure. Indeed, subgraphs of **Tuple+Clique** only composed of tuples are very dense and none of the nodes constituting these subgraphs are deleted by the perturbation. The counterpart to this high density graph is a much less efficient learning processing. In comparison **5-DB Gaif** although it too has cliques is twice sparser (as shown in Table 1).

## 7 CONCLUSION

Motivated by the dynamic nature of databases, this research investigates the stability properties of embeddings generated by an important class of GNNs, Signal Processing GNNs (SPGNNs). We refine existing global stability bounds by eliminating the dependence on graph size. Additionally, we extend our analysis to the node level. When applied to graphs generated from relational databases, we demonstrate that considering the heterogeneous and heterophilic nature of these graphs can lead to more stable embeddings. To achieve this, we propose improved database-to-graph constructions that outperform existing methods in terms of embedding stability.

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

# A  STABILITY

## A.1  PROOF OF THEOREM 2

We follow a method that is similar to what is done in Gama et al. (2020).

We define $||H_{f,g}^{(l)}||_\infty := \max\limits_{\lambda \in \sigma(\boldsymbol{S})} |H_{f,g}^{(l)}(\lambda)|$ and $||\hat{H}_{f,g}^{(l)}||_\infty := \max\limits_{\lambda \in \sigma(\hat{\boldsymbol{S}})} |H_{f,g}^{(l)}(\lambda)|$.

We will use regularly the fact that $||H_{f,g}^{(l)}||_\infty = ||H_{f,g}^{(l)}(\boldsymbol{S})||_{op}$ and $||\hat{H}_{f,g}^{(l)}||_\infty = ||H_{f,g}^{(l)}(\hat{\boldsymbol{S}})||_{op}$, which follows from the definition.

We begin with a lemma:

**Lemma 3.** We consider a SPGNN $\Phi$ satisfying the hypotheses of Theorem 2, $\hat{\boldsymbol{S}}$ a symmetric matrix and $\boldsymbol{x}$ a vector. Then we have:

$$||\Phi(\hat{\boldsymbol{S}}, \boldsymbol{x})|| \leq \left( \sum_{\boldsymbol{f} \in Lay^{(L)}} \prod_{l=0}^{L-1} ||\hat{H}_{\boldsymbol{f}_l, \boldsymbol{f}_{l+1}}^{(l)}||_\infty \right) ||\boldsymbol{x}||$$

where $Lay^{(L)} := \{1, \ldots, F^{(0)}\} \times \{1, \ldots, F^{(1)}\} \times \cdots \times \{1, \ldots, F^{(L)}\}$.

*Proof.* Indeed, we show this lemma by induction on the number of layers of $\Phi$. For a SPGNN $\Phi$ with only one layer, since $F^{(0)} = F^{(1)} = 1$ by definition, then $\Phi$ is of the form $\Phi(\hat{\boldsymbol{S}}, \boldsymbol{x}) = \sigma(H_{1,1}^{(0)}(\hat{\boldsymbol{S}})\boldsymbol{x})$. Consequently,

$$||\Phi(\hat{\boldsymbol{S}}, \boldsymbol{x})|| \leq ||H_{1,1}^{(0)}(\hat{\boldsymbol{S}})\boldsymbol{x}||$$
$$\leq ||\hat{H}_{1,1}^{(0)}||_\infty ||\boldsymbol{x}||.$$

This proves that the lemma holds for the initial case.

We now consider $L \in \mathbb{N}^*$ and we assume that the lemma is true for all SPGNNs with $L$ layers satisfying the hypotheses.

Let $\Phi(\cdot, \cdot)$ be a SPGNN with $L + 1$ layers satisfying the hypotheses, let $\hat{\boldsymbol{S}}$ be a symmetric matrix and let $\boldsymbol{x}$ be a vector. By definition of the SPGNNs, $\Phi(\hat{\boldsymbol{S}}, \boldsymbol{x})$ has the form $\Phi(\hat{\boldsymbol{S}}, \boldsymbol{x}) = \sigma(\sum_{f^{(L)}=1}^{F^{(L)}} H_{f^{(L)},1}^{(L)}(\hat{\boldsymbol{S}})\Phi_{f^{(L)}}^{(L)}(\hat{\boldsymbol{S}}, \boldsymbol{x}))$ where $(\Phi_f^{(L)})$ are SPGNNs themselves, with only $L$ layers. When applying the induction hypothesis to SPGNNs with $L$ layers, we use the notation $Lay^{(L,f^{(L)})}$ to define the set of vectors $\boldsymbol{f} \in Lay^{(L)}$ whose last coordinate is $\boldsymbol{f}_L = f^{(L)}$ rather than $\boldsymbol{f}_L = 1$. (The last coordinate of vectors in $Lay^{(L)}$ is always 1 since $F^{(L)} = 1$ in a $L$-layer SPGNN.)

We can then write:

$$||\Phi(\hat{\boldsymbol{S}}, \boldsymbol{x})|| \leq \sum_{f^{(L)}=1}^{F^{(L)}} ||\hat{H}_{f^{(L)},1}^{(L)}||_\infty ||\Phi_{f^{(L)}}^{(L)}(\hat{\boldsymbol{S}}, \boldsymbol{x})||$$

$$\leq \sum_{f^{(L)}=1}^{F^{(L)}} \sum_{\boldsymbol{f} \in Lay^{(L,f^{(L)})}} ||\hat{H}_{f^{(L)},1}^{(L)}||_\infty \left( \prod_{l=0}^{L-1} ||\hat{H}_{\boldsymbol{f}_l, \boldsymbol{f}_{l+1}}^{(l)}||_\infty \right) ||\boldsymbol{x}||,$$

hence $||\Phi(\hat{\boldsymbol{S}}, \boldsymbol{x})|| \leq \sum_{\boldsymbol{f} \in Lay^{(L+1)}} \prod_{l=0}^{L} ||\hat{H}^{(l)}_{\boldsymbol{f}_{l+1}, \boldsymbol{f}_l}||_{\infty} ||\boldsymbol{x}||.$

This proves the lemma. $\qquad\square$

*Proof.* We now prove Theorem 2 by induction on the number of layers $L$. We prove a more precise bound:

$$||\Phi(\boldsymbol{S}, \boldsymbol{x}) - \Phi(\hat{\boldsymbol{S}}, \boldsymbol{x})|| \leq B^{(L)} ||\boldsymbol{S} - \hat{\boldsymbol{S}}||_{op} ||\boldsymbol{x}||$$

where $B^{(L)} := \sum_{\boldsymbol{f} \in Lay^{(L)}} \sum_{k=1}^{L} \left[ ( \prod_{l=L-k+1}^{L-1} ||H^{(l)}_{\boldsymbol{f}_l, \boldsymbol{f}_{l+1}}||_{\infty}) \cdot A^{(L-k)}_{\boldsymbol{f}_{L-k}, \boldsymbol{f}_{L-k+1}} \cdot ( \prod_{l=0}^{L-k-1} ||\hat{H}^{(l)}_{\boldsymbol{f}_l, \boldsymbol{f}_{l+1}}||_{\infty}) \right].$

We remind that $Lay^{(L)} = \{1, \ldots, F^{(0)}\} \times \cdots \times \{1, \ldots, F^{(L)}\}$, $||H^{(l)}_{f,g}||_{\infty} := \max_{\lambda \in \sigma(\boldsymbol{S})} |H^{(l)}(\lambda)|$ and $A^{(l)f,g}$ is defined as $A^{(l)f,g} = |a|$ where $a$ is the coefficient such that $H^{(l)}_{f,g}(X) = aX + b$. This bound is much less readable, but it is more accurate, because it does not assume that the different parameters are uniform in the SPGNN.

To motivate the technical proof below, we begin by showing that the result shown ($||\Phi(\boldsymbol{S}, \boldsymbol{x}) - \Phi(\hat{\boldsymbol{S}}, \boldsymbol{x})|| \leq B^{(L)} ||\boldsymbol{S} - \hat{\boldsymbol{S}}||_{op} ||\boldsymbol{x}||$) implies Theorem 2, which means that $B^{(L)}$ is indeed a tighter bound.

To achieve this objective, it suffices to show that $B^{(L)} \leq LA(F||H||_{\infty})^{L-1}$. This can be seen by replacing all the terms that appears in $B^{(L)}$ by their global counterpart:

$$B^{(L)} = \sum_{\boldsymbol{f} \in Lay^{(L)}} \sum_{k=1}^{L} \left[ ( \prod_{l=L-k+1}^{L-1} ||H^{(l)}_{\boldsymbol{f}_l, \boldsymbol{f}_{l+1}}||_{\infty}) \cdot A^{(L-k)}_{\boldsymbol{f}_{L-k}, \boldsymbol{f}_{L-k+1}} \cdot ( \prod_{l=0}^{L-k-1} ||\hat{H}^{(l)}_{\boldsymbol{f}_l, \boldsymbol{f}_{l+1}}||_{\infty}) \right]$$

$$\leq \sum_{\boldsymbol{f} \in Lay^{(L)}} \sum_{k=1}^{L} \left[ ( \prod_{l=L-k+1}^{L-1} ||H||_{\infty}) \cdot A \cdot ( \prod_{l=0}^{L-k-1} ||H||_{\infty}) \right]$$

$$= A(||H||_{\infty})^{L-1} \sum_{\boldsymbol{f} \in Lay^{(L)}} \sum_{k=1}^{L} 1$$

$$= A(||H||_{\infty})^{L-1} L |Lay^{(L)}|$$

$$= A(||H||_{\infty})^{L-1} L \prod_{l=0}^{L} F^{(l)}$$

$$\leq A(||H||_{\infty})^{L-1} L F^{L-1}.$$

We remind that $F^{(0)} = F^{(L)} = 1$, which is explains why $F^{L-1}$ appears in the last inequality rather than $F^{L+1}$.

We now prove the base case of the induction. We consider a GNN with 1 layer $\Phi(\boldsymbol{S}, \boldsymbol{x}) = \sigma(H(\boldsymbol{S})\boldsymbol{x})$ with $H(\boldsymbol{S}) = a\boldsymbol{S} + b\boldsymbol{I}_N$. Given two symmetric matrices $\boldsymbol{S}$ and $\hat{\boldsymbol{S}}$ and a vector $\boldsymbol{x} \in \mathbb{R}^N$, the theorem is written: $||\sigma(H(\boldsymbol{S})\boldsymbol{x}) - \sigma(H(\hat{\boldsymbol{S}})\boldsymbol{x})|| \leq B^{(1)} ||S - \hat{S}||_{op} ||x||$ with $B^{(1)} = A^{(1)}_{1,1} = |a|$.

This holds since:

$$||\sigma(H(\boldsymbol{S})\boldsymbol{x}) - \sigma(H(\hat{\boldsymbol{S}})\boldsymbol{x})|| \leq ||H(\boldsymbol{S})\boldsymbol{x} - H(\hat{\boldsymbol{S}})\boldsymbol{x}||$$

$$= ||a(\boldsymbol{S} - \hat{\boldsymbol{S}})\boldsymbol{x}||$$

$$\leq |a| ||\boldsymbol{S} - \hat{\boldsymbol{S}}||_{op} ||\boldsymbol{x}||.$$

We now consider $L \in \mathbb{N}^*$ and we assume that the theorem holds for SPGNNs with $L$ layers.

We consider a SPGNN $\Phi$ with $L + 1$ layers and we use the exact same notations as in the proof of Lemma 3, i.e. $\Phi(\boldsymbol{S}, \boldsymbol{x}) = \sigma \left( \sum_{f^{(L)}}^{F^{(L)}} H^{(L)}_{f^{(L)}, 1}(S) \Phi^{(L)}_{f^{(L)}}(\boldsymbol{S}, \boldsymbol{x}) \right).$

We begin by bounding $\Phi(\boldsymbol{S}, \boldsymbol{x}) - \Phi(\hat{\boldsymbol{S}}, \boldsymbol{x})$ with the $\Phi_i^{(L)}$:

$$||\Phi(\boldsymbol{S}, \boldsymbol{x}) - \Phi(\hat{\boldsymbol{S}}, \boldsymbol{x})|| = ||\sigma\left(\sum_{f^{(L)}=1}^{F^{(L)}} H_{f^{(L)},1}^{(L)}(\boldsymbol{S})\Phi_{f^{(L)}}^{(L)}(\boldsymbol{S}, \boldsymbol{x})\right) - \sigma\left(\sum_{f^{(L)}=1}^{F^{(L)}} H_{f^{(L)},1}^{(L)}(\hat{\boldsymbol{S}})\Phi_{f^{(L)}}^{(L)}(\hat{\boldsymbol{S}}, \boldsymbol{x})\right)||$$

$$\leq \sum_{f^{(L)}=1}^{F^{(L)}} ||H_{f^{(L)},1}^{(L)}(\boldsymbol{S})\Phi_{f^{(L)}}^{(L)}(\boldsymbol{S}, \boldsymbol{x}) - H_{f^{(L)},1}^{(L)}(\hat{\boldsymbol{S}})\Phi_{f^{(L)}}^{(L)}(\hat{\boldsymbol{S}}, \boldsymbol{x})||$$

$$\leq \sum_{f^{(L)}=1}^{F^{(L)}} ||H_{f^{(L)},1}^{(L)}(\boldsymbol{S})\left(\Phi_{f^{(L)}}^{(L)}(\boldsymbol{S}, \boldsymbol{x}) - \Phi_{f^{(L)}}^{(L)}(\hat{\boldsymbol{S}}, \boldsymbol{x})\right)||$$

$$+ \sum_{f^{(L)}=1}^{F^{(L)}} ||\left(H_{f^{(L)},1}^{(L)}(\boldsymbol{S}) - H_{f^{(L)},1}^{(L)}(\hat{\boldsymbol{S}})\right)\Phi_{f^{(L)}}^{(L)}(\hat{\boldsymbol{S}}, \boldsymbol{x})|| \quad (5)$$

We should now bound both terms in inequality 5 separately. We begin with $||H_{f^{(L)},1}^{(L)}(\boldsymbol{S})\left(\Phi_{f^{(L)}}^{(L)}(\boldsymbol{S}, \boldsymbol{x}) - \Phi_{f^{(L)}}^{(L)}(\hat{\boldsymbol{S}}, \boldsymbol{x})\right)||$:

$$||H_{f^{(L)},1}^{(L)}(\boldsymbol{S})\left(\Phi_{f^{(L)}}^{(L)}(\boldsymbol{S}, \boldsymbol{x}) - \Phi_{f^{(L)}}^{(L)}(\hat{\boldsymbol{S}}, \boldsymbol{x})\right)||$$

$$\leq ||H_{f^{(L)},1}^{(L)}||_\infty ||\Phi_{f^{(L)}}^{(L)}(\boldsymbol{S}, \boldsymbol{x}) - \Phi_{f^{(L)}}^{(L)}(\hat{\boldsymbol{S}}, \boldsymbol{x})||$$

$$\leq ||H_{f^{(L)},1}^{(L)}||_\infty B_{f^{(L)}}^{(L)}||\boldsymbol{S} - \hat{\boldsymbol{S}}||_{op}||\boldsymbol{x}|| \quad (6)$$

where $B_{f^{(L)}}^{(L)}$ is the bound of the theorem applied to the $L$-layer SPGNN $\Phi_{f^{(L)}}^L$.

For the second term, using the definition of $A^{(L)}$ and Lemma 3, we get:

$$||\left(H_{f^{(L)},1}^{(L)}(\boldsymbol{S}) - H_{f^{(L)},1}^{(L)}(\hat{\boldsymbol{S}})\right)\Phi_{f^{(L)}}^{(L)}(\hat{\boldsymbol{S}}, \boldsymbol{x})||$$

$$\leq A_{f^{(L)},1}^{(L)}||\boldsymbol{S} - \hat{\boldsymbol{S}}||_{op}||\Phi_{f^{(L)}}^{(L)}(\hat{\boldsymbol{S}}, \boldsymbol{x})||$$

$$\leq A_{f^{(L)},1}^{(L)}||\boldsymbol{S} - \hat{\boldsymbol{S}}||_{op}\left(\sum_{\boldsymbol{f}\in Lay^{(L)}}\prod_{l=0}^{L-1}||\hat{H}_{\boldsymbol{f}_l,\boldsymbol{f}_{l+1}}^{(l)}||_\infty\right)||\boldsymbol{x}||. \quad (7)$$

Combining the bounds given by inequalities 6 and 7 in the inequality 5 yields

$$||\Phi(\boldsymbol{S}, \boldsymbol{x}) - \Phi(\hat{\boldsymbol{S}}, \boldsymbol{x})|| \quad (8)$$

$$\leq \sum_{f^{(L)}=1}^{F^{(L)}} ||H_{f^{(L)},1}^{(L)}||_\infty B_{f^{(L)}}^{(L)}||\boldsymbol{S} - \hat{\boldsymbol{S}}||_{op}||\boldsymbol{x}||$$

$$+ \sum_{f^{(L)}=1}^{F^{(L)}} A_{f^{(L)},1}^{(L)}\left(\sum_{\boldsymbol{f}\in Lay^{(L)}}\prod_{l=0}^{L-1}||\hat{H}_{\boldsymbol{f}_l,\boldsymbol{f}_{l+1}}^{(l)}||_\infty\right)||\boldsymbol{S} - \hat{\boldsymbol{S}}||_{op}||\boldsymbol{x}|| \quad (9)$$

$$= \sum_{f^{(L)}=1}^{F^{(L)}} \left(||H_{f^{(L)},1}^{(L)}||_\infty B_{f^{(L)}}^{(L)} + A_{f^{(L)},1}^{(L)}\left(\sum_{\boldsymbol{f}\in Lay^{(L)}}\prod_{l=0}^{L-1}||\hat{H}_{\boldsymbol{f}_l,\boldsymbol{f}_{l+1}}^{(l)}||_\infty\right)\right)||\boldsymbol{S} - \hat{\boldsymbol{S}}||_{op}||\boldsymbol{x}||.$$

$$(10)$$

Given that $\sum_{f^{(L)}=1}^{F^{(L)}} \left(||H_{f^{(L)},1}^{(L)}||_\infty B_{f^{(L)}}^{(L)} + A_{f^{(L)},1}^{(L)}\left(\sum_{\boldsymbol{f}\in Lay^{(L)}}\prod_{l=0}^{L-1}||\hat{H}_{\boldsymbol{f}_l,\boldsymbol{f}_{l+1}}^{(l)}||_\infty\right)\right) = B^{(L+1)}$, this can be rewritten:

$$||\Phi(\boldsymbol{S}, \boldsymbol{x}) - \Phi(\hat{\boldsymbol{S}}, \boldsymbol{x})|| \leq B^{(L+1)} ||\boldsymbol{S} - \hat{\boldsymbol{S}}||_{op} ||\boldsymbol{x}||, \tag{11}$$

which shows that the theorem holds for GNN with $L + 1$ layers and concludes the proof.

□

We would like to highlight that the proof only relies on on the fact $||H_{i,j}^{(l)}(\boldsymbol{S})||_{op} = ||H_{i,j}^{(l)}||_{\infty}$ (through Lemma 3 and inequality 6) and on the bound $||H_{i,j}^{(l)}(\boldsymbol{S}) - H_{i,j}^{(l)}(\hat{\boldsymbol{S}})||_{op} \leq A_{i,j}^{(l)} ||\boldsymbol{S} - \hat{\boldsymbol{S}}||_{op}$ (through the base case of the induction and inequality 7). The rest of the proof only involves general computations that do not require any specific hypothesis, and the complicated expression of $B^{(L)}$ is simply what appears to be the tightest bound that our method can yield. In particular, if we have different bounds for $||H_{i,j}^{(l)}(\boldsymbol{S})||_{op}$ and $||H_{i,j}^{(l)}(\boldsymbol{S}) - H_{i,j}^{(l)}(\hat{\boldsymbol{S}})||_{op}$ in other settings, then Theorem 2 applies if we replace the $||H_{i,j}^{(l)}||_{\infty}$ and $A_{i,j}^{(l)}$ in the definition of $B^{(L)}$ by those bounds. This observation is the ground for two important generalizations of Theorem 2.

The first generalization allows to apply the theorem to polynomials of higher order and is based on the fact that we can provide a bound for $||(H_{f^{(L)},1}^{(L)}(\boldsymbol{S}) - H_{f^{(L)},1}^{(L)}||_{op}$ even when $H_{f^{(L)},1}^{(L)}$ is *not* an order one polynomial, while the definition $||H_{i,j}^{(l)}||_{\infty} := \max_{\lambda \in \sigma(\boldsymbol{S})} |H_{i,j}^{(l)}(\lambda)|$ is still equal to $||H_{i,j}^{(l)}(\boldsymbol{S})||_{op}$ independently of the degree of $H$. This is detailed in Appendix A.4.1.

The second generalization allows to apply the theorem to non-symmetric GSOs. The main point is that the symmetry of the GSOs $\boldsymbol{S}$ is only used to efficiently compute the operator norm of $H(\boldsymbol{S})$ through $||H||_{\infty} = \max_{\lambda \in \sigma(\boldsymbol{S})} |H(\lambda)|$ (when $H$ is a degree 1 polynomial, it is enough to evaluate this expression on the smallest and largest eigenvalues of $\boldsymbol{S}$, which is $O(1)$ operations). However, replacing $||H||_{\infty}$ by $||H(S)||_{op}$ is perfectly rigorous. The only issue is the computational cost of repeated computation of operator norms for non-symmetric matrices, but we can solve this issue by loosening the bound using the inequality, if $H(X) = \sum_{i=0}^{d} h_i X^i$ is a filter, $||H(\boldsymbol{S})||_{op} \leq \sum_{i=0}^{d} |h_i| ||\boldsymbol{S}||_{op}^i$. This means that it is enough to compute the operator of a $N \times N$ matrix once, and it is not necessary to do it for every filters. This is further detailed in Appendix A.4.2.

## A.2 PROOFS OF THE COROLLARIES.

We begin by proving Corollary 2.1:

*Proof.* We consider $\Phi$, $\boldsymbol{S}$, $\hat{\boldsymbol{S}}$ and $\boldsymbol{X}^{(0)}$ as in the statement of the corollary.

By definition of the embedding, $||\Phi(\boldsymbol{S}, \boldsymbol{X}^{(0)})_{v_i} - \Phi(\hat{\boldsymbol{S}}, \boldsymbol{X}^{(0)})_{\hat{v}_i}|| = ||L_i||^2$ where $L_i$ is the $i$-th line of the matrix $\Phi(\boldsymbol{S}, \boldsymbol{X}^{(0)}) - \Phi(\hat{\boldsymbol{S}}, \boldsymbol{X}^{(0)})$. Therefore,

$$\sum_{i=1}^{N} ||\Phi(\boldsymbol{S}, \boldsymbol{X}^{(0)})_{v_i} - \Phi(\hat{\boldsymbol{S}}, \boldsymbol{X}^{(0)})_{v'_i}||^2 = \sum_{i=1}^{N} ||L_i||^2$$
$$= \sum_{j=1}^{d} ||C_j||^2$$
$$= \sum_{j=1}^{d} ||\Phi(\boldsymbol{S}, \boldsymbol{X}_{:,j}^{(0)}) - \Phi(\hat{\boldsymbol{S}}, \boldsymbol{X}_{:,j}^{(0)})||^2$$

where $C_j$ is the $j$-th column of the matrix $\Phi(\boldsymbol{S}, \boldsymbol{X}^{(0)}) - \Phi(\hat{\boldsymbol{S}}, \boldsymbol{X}^{(0)})$.

Since Theorem 2 ensures that $||\Phi(\boldsymbol{S}, C_j^{(0)}) - \Phi(\hat{\boldsymbol{S}}, C_j^{(0)})|| \leq M$, we deduce:

$$\sum_{i=1}^{N} ||\Phi(\boldsymbol{S}, \boldsymbol{X}^{(0)})_{v_i} - \Phi(\hat{\boldsymbol{S}}, \boldsymbol{X}^{(0)})_{\hat{v}_i}||^2 \leq dM^2.$$

$\square$

We now prove Corollary 2.2. Similarly as what is done for the proof of Theorem 2, we provide a tighter bound: $||\Phi(\boldsymbol{S}, \boldsymbol{X}^{(0)})_{v_i} - \Phi(\hat{\boldsymbol{S}}, \boldsymbol{X}^{(0)})_{v'_i}|| \leq \sqrt{d}B_k^{(L)}||\boldsymbol{S} - \hat{\boldsymbol{S}}||_{op}$ where

$$B_k^{(L)} := \sum_{\boldsymbol{f} \in Lay^{(L)}} \sum_{m=k+1}^{L} \left[ \left( \prod_{l=L-m+1}^{L-1} ||H_{\boldsymbol{f}_l, \boldsymbol{f}_{l+1}}^{(l)}||_\infty \right) \cdot A_{\boldsymbol{f}_{L-m}, \boldsymbol{f}_{L-m+1}}^{(L-m)} \cdot \left( \prod_{l=0}^{L-m-1} ||\hat{H}_{\boldsymbol{f}_l, \boldsymbol{f}_{l+1}}^{(l)}||_\infty \right) \right].$$

*Proof.* For the purpose of this proof, we introduce SPGNNs that are slightly more generals than the SPGNNs used previously: we allow the SPGNN to have $F^{(0)} \geq 1$; i.e. they take $F^{(0)}$ vectors as input rather than one, and we note $\Phi(\boldsymbol{S}, (\boldsymbol{x}_1, \boldsymbol{x}_2, \ldots, \boldsymbol{x}_{F^{(0)}}))$ such a generalized SPGNN with initialization vectors $(\boldsymbol{x}_1, \boldsymbol{x}_2, \ldots, \boldsymbol{x}_{F^{(0)}})$.

We consider a generalized SPGNN $\Phi$ with $L$ layers, and a couple of identified nodes $v_i, \hat{v}_i \in \mathcal{G}, \hat{\mathcal{G}}$ having their $k$-hop equal. We prove by induction on $L$ that, if $k \geq L$, then $\Phi(\boldsymbol{S}, (\boldsymbol{x}_1, \boldsymbol{x}_2, \ldots, \boldsymbol{x}_{F^{(0)}}))_i = \Phi(\hat{S}, (\boldsymbol{x}_1, \boldsymbol{x}_2, \ldots, \boldsymbol{x}_{F^{(0)}}))_i$.

If $L = 0$ then $\Phi(\boldsymbol{S}, (\boldsymbol{x}_1, \boldsymbol{x}_2, \ldots, \boldsymbol{x}_{F^{(0)}})) = (\boldsymbol{x}_1, \boldsymbol{x}_2, \ldots, \boldsymbol{x}_{F^{(0)}})$ and the property is true.

We now consider $L \in \mathbb{N}^*$ and we assume that the theorem holds for SPGNNs with $L$ layers. We consider a SPGNN $\Phi$ with $L+1$ layers and we use the exact same notations as in the previous proofs, i.e. $\Phi(\boldsymbol{S}, (\boldsymbol{x}_1, \boldsymbol{x}_2, \ldots, \boldsymbol{x}_{F^{(0)}})) = \sigma \left( \sum_{f^{(L)}}^{F^{(L)}} H_{f^{(L)},1}^{(L)}(S) \Phi_{f^{(L)}}^{(L)}(\boldsymbol{S}, (\boldsymbol{x}_1, \boldsymbol{x}_2, \ldots, \boldsymbol{x}_{F^{(0)}})) \right)$.

Since $\sigma$ is point-wise, we have:

$$\Phi(\boldsymbol{S}, (\boldsymbol{x}_1, \boldsymbol{x}_2, \ldots, \boldsymbol{x}_{F^{(0)}}))_i = \sigma \left( \sum_{f^{(L)}}^{F^{(L)}} \left( H_{f^{(L)},1}^{(L)}(S) \Phi_{f^{(L)}}^{(L)}(\boldsymbol{S}, (\boldsymbol{x}_1, \boldsymbol{x}_2, \ldots, \boldsymbol{x}_{F^{(0)}})) \right)_i \right)$$

$$= \sigma \left( \sum_{f^{(L)}}^{F^{(L)}} \sum_{k=1}^{N} (H_{f^{(L)},1}^{(L)})_{i,k}(S)(\Phi_{f^{(L)}}^{(L)}(\boldsymbol{S}, (\boldsymbol{x}_1, \boldsymbol{x}_2, \ldots, \boldsymbol{x}_{F^{(0)}})))_k \right).$$

However, in the last sum $(H_{f^{(L)},1}^{(L)})_{i,k}(S) = 0$ if $k$ is not the index of a neighbour of $v_i$ in $\mathcal{G}$. Therefore, we can apply the induction hypothesis on the terms $(\Phi_{f^{(L)}}^{(L)}(\boldsymbol{S}, (\boldsymbol{x}_1, \boldsymbol{x}_2, \ldots, \boldsymbol{x}_{F^{(0)}})))_k$ because the neighbour of $v_i$ have equal $k-1$-hops, and $(H_{f^{(L)},1}^{(L)})_{i,k}(S) = (H_{f^{(L)},1}^{(L)})_{i,k}(\hat{S})$ by definition of the $k$-hop equality.

This yields:

$$\Phi(\boldsymbol{S}, (\boldsymbol{x}_1, \boldsymbol{x}_2, \ldots, \boldsymbol{x}_{F^{(0)}}))_i = \sigma \left( \sum_{f^{(L)}}^{F^{(L)}} \left( H_{f^{(L)},1}^{(L)}(S) \Phi_{f^{(L)}}^{(L)}(\boldsymbol{S}, (\boldsymbol{x}_1, \boldsymbol{x}_2, \ldots, \boldsymbol{x}_{F^{(0)}})) \right)_i \right)$$

$$= \sigma \left( \sum_{f^{(L)}}^{F^{(L)}} \left( H_{f^{(L)},1}^{(L)}(\hat{S}) \Phi_{f^{(L)}}^{(L)}(\hat{\boldsymbol{S}}, (\boldsymbol{x}_1, \boldsymbol{x}_2, \ldots, \boldsymbol{x}_{F^{(0)}})) \right)_i \right)$$

$$= \Phi(\hat{S}, (\boldsymbol{x}_1, \boldsymbol{x}_2, \ldots, \boldsymbol{x}_{F^{(0)}}))_i.$$

We now use this result to show the corollary. Firstly, we remark that we can replace the initialisation matrix $\boldsymbol{X}^{(0)}$ by a vector $\boldsymbol{x}$ (i.e. consider embeddings in dimension $d = 1$) since going from $d = 1$ to a general $d \in \mathbb{N}$ only involves applying the inequality to each element of the vector $\Phi(\boldsymbol{S}, \boldsymbol{X}^{(0)}) - \Phi(\hat{\boldsymbol{S}}, \boldsymbol{X}^{(0)})$, similarly than what is done in the proof of Corollary 2.1. Note that, in this setting, $\Phi(\boldsymbol{S}, \boldsymbol{x})_{v_i}$ is the $i$-th coordinate of the vector $\Phi(\boldsymbol{S}, \boldsymbol{x})$, noted $\Phi(\boldsymbol{S}, \boldsymbol{x})_i$.

Then, we remark that $\Phi(\boldsymbol{S}, \boldsymbol{x}) = \Phi^{(L-k),(L)}(\boldsymbol{S}, (\Phi_1^{(L-K)}(\boldsymbol{S}, \boldsymbol{x}), \Phi_2^{(L-K)}(\boldsymbol{S}, \boldsymbol{x}), \ldots, \Phi_{F^{(L-k)}}^{(L-K)}(\boldsymbol{S}, \boldsymbol{x})))$ where $\Phi_f^{(L-K)}$ is the sub-SPGNN of $\Phi$ that returns the intermediate value of $\Phi$ in the $f$-th com-

ponents of the $(L-k)$-th layer, and $\Phi^{(L-k),(L)}$ is the generalized SPGNN that transforms the $(L-k)$-th layer in the last one.

By the previous result, we have

$$(\Phi(\boldsymbol{S}, \boldsymbol{X}^{(0)}) - \Phi(\hat{\boldsymbol{S}}, \boldsymbol{X}^{(0)}))_i =$$

$$(\Phi^{(L-k),(L)}(\boldsymbol{S}, (\Phi_1^{(L-K)}(\boldsymbol{S}, \boldsymbol{x}), \Phi_2^{(L-K)}(\boldsymbol{S}, \boldsymbol{x}), \dots, \Phi_{F(L-k)}^{(L-K)}(\boldsymbol{S}, \boldsymbol{x}))))_i$$

$$- (\Phi^{(L-k),(L)}(\boldsymbol{S}, (\Phi_1^{(L-K)}(\hat{\boldsymbol{S}}, \boldsymbol{x}), \Phi_2^{(L-K)}(\hat{\boldsymbol{S}}, \boldsymbol{x}), \dots, \Phi_{F(L-k)}^{(L-K)}(\hat{\boldsymbol{S}}, \boldsymbol{x}))))_i.$$

Finally, the proof of Theorem 2 shows that this vector can be bounded by $B_k^{(L)} ||\boldsymbol{S} - \hat{\boldsymbol{S}}||_{op} ||\boldsymbol{x}||$. (Basically because, in the induction step, the term $|| (H_{f^{(L)},1}^{(L)}(\boldsymbol{S}) - H_{f^{(L)},1}^{(L)}(\hat{\boldsymbol{S}}))\Phi_{f^{(L)}}^{(L)}(\hat{\boldsymbol{S}}, \boldsymbol{x}) ||$, which is the term studied in 7, is 0 for the last $k$ layers.) $\square$

### A.3 INSTANTIATION IN THE DISTANCE MODULO PERMUTATION FRAMEWORK

We remind the *distance modulo permutation* framework, that is commonly used to give stability results: To measure the distance between two graph shift operators, and consequently the distance between the graphs they are derived from, we use $d_{\mathcal{P}}(\boldsymbol{S}, \hat{\boldsymbol{S}})$, denoted $||\boldsymbol{S} - \hat{\boldsymbol{S}}||_{\mathcal{P}}$ in Gama et al. (2020), defined as $d_{\mathcal{P}}(\boldsymbol{S}, \hat{\boldsymbol{S}}) = \min_{\boldsymbol{P} \in \mathcal{P}} ||\boldsymbol{S} - \boldsymbol{P}\hat{\boldsymbol{S}}\boldsymbol{P}^T||$ where $\mathcal{P}$ is the set of permutation matrices. We change the notations to highlight the fact that the quantity depends on the two matrices $\boldsymbol{S}, \hat{\boldsymbol{S}}$ and not on the difference $\boldsymbol{S} - \hat{\boldsymbol{S}}$. The definition of $d_{\mathcal{P}}$ can be extended to general continuous operators $\Psi, \hat{\Psi} : \mathbb{R}^N \to \mathbb{R}^N$ by $d_{\mathcal{P}}(\Psi, \hat{\Psi}) = \min_{\boldsymbol{P} \in \mathcal{P}} \max_{||x||=1} ||\boldsymbol{P}^T \Psi(x) - \hat{\Psi}(\boldsymbol{P}^T x)||$.

We can now give a version of Theorem 2 in this framework:

**Corollary 3.1.** Given a GNN $\Phi$ satisfying the hypotheses of Theorem 2, the following equality holds:
$$d_{\mathcal{P}}(\Phi(\boldsymbol{S}, \cdot), \Phi(\hat{\boldsymbol{S}}, \cdot)) \leq LA(F||H||_\infty)^{L-1} d_{\mathcal{P}}(\boldsymbol{S}, \hat{\boldsymbol{S}}).$$

*Proof.* We consider $\boldsymbol{P}_0 \in \mathcal{P}$ such that $d_{\mathcal{P}}(\boldsymbol{S}, \hat{\boldsymbol{S}}) = ||\boldsymbol{S} - \boldsymbol{P}_0 \hat{\boldsymbol{S}} \boldsymbol{P}_0^T||_{op}$. The theorem then provides the inequality, for $||\boldsymbol{x}|| = 1$,

$$||\Phi(\boldsymbol{S}, \boldsymbol{x}) - \Phi(\boldsymbol{P}_0 \hat{\boldsymbol{S}} \boldsymbol{P}_0^T, \boldsymbol{x})|| \leq LA(F||H||_\infty)^{L-1} ||\boldsymbol{S} - \boldsymbol{P}_0 \hat{\boldsymbol{S}} \boldsymbol{P}_0^T||_{op}$$
$$= LA(F||H||_\infty)^{L-1} d_{\mathcal{P}}(\boldsymbol{S}, \hat{\boldsymbol{S}})$$

and since

$$d_{\mathcal{P}}(\Phi(\boldsymbol{S}, \cdot), \Phi(\hat{\boldsymbol{S}}, \cdot)) = \min_{\boldsymbol{P} \in \mathcal{P}} \max_{||\boldsymbol{x}||=1} ||\boldsymbol{P}^T \Phi(\boldsymbol{S}, \boldsymbol{x}) - \Phi(\hat{\boldsymbol{S}}, \boldsymbol{P}^T \boldsymbol{x})||$$
$$= \min_{\boldsymbol{P} \in \mathcal{P}} \max_{||\boldsymbol{x}||=1} ||\Phi(\boldsymbol{S}, \boldsymbol{x}) - \boldsymbol{P}\Phi(\hat{\boldsymbol{S}}, \boldsymbol{P}^T \boldsymbol{x})||$$
$$= \min_{\boldsymbol{P} \in \mathcal{P}} \max_{||\boldsymbol{x}||=1} ||\Phi(\boldsymbol{S}, \boldsymbol{x}) - \Phi(\boldsymbol{P}\hat{\boldsymbol{S}}\boldsymbol{P}^T, \boldsymbol{x})||$$
$$\leq \max_{||\boldsymbol{x}||=1} ||\Phi(\boldsymbol{S}, \boldsymbol{x}) - \Phi(\boldsymbol{P}_0 \hat{\boldsymbol{S}} \boldsymbol{P}_0^T, \boldsymbol{x})||,$$

then we have for the $\boldsymbol{x}$ that reaches the max:

$$d_{\mathcal{P}}(\Phi(\boldsymbol{S}, \cdot), \Phi(\hat{\boldsymbol{S}}, \cdot)) \leq ||\Phi(\boldsymbol{S}, \boldsymbol{x}) - \Phi(\boldsymbol{P}\hat{\boldsymbol{S}}\boldsymbol{P}^T, \boldsymbol{x})||$$
$$\leq LA(F||H||_\infty)^{L-1} d_{\mathcal{P}}(\boldsymbol{S}, \hat{\boldsymbol{S}}).$$

This proves the corollary.

$\square$

The same instantiation can be made for Corollaries 2.1 and 2.2.

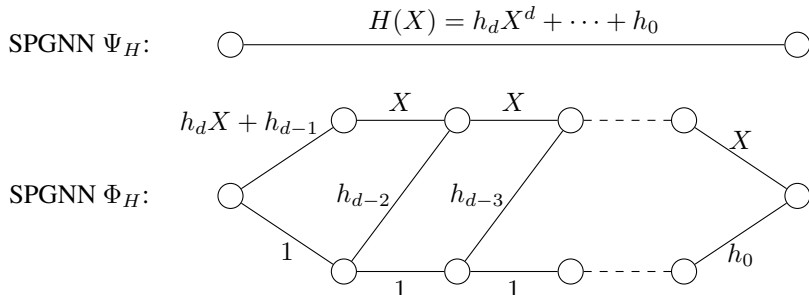

Figure 4: Construction of a SPGNN $\Phi_H$ that only contains one order polynomials from a SPGNN $\Psi_H$ that contains a polynomial of order $d$.

### A.4 GENERALIZATION OF THEOREM 2.

As mentioned at the end of the proof of Theorem 2, we can generalize the result to SPGNNs that use polynomials of any degree and non-symmetric GSOs.

#### A.4.1 HYPOTHESIS ON THE DEGREE OF THE FILTERS

We begin by showing that if we consider a SPGNN $\Psi_H$ of the form $\Psi_H(\boldsymbol{S}, \boldsymbol{x}) = H(\boldsymbol{S})\boldsymbol{x}$ where $H$ is a polynomial of any degree, then there exists a SPGNN $\Phi_H$ that uses only polynomials of order one and satisfies $\Psi_H(\boldsymbol{S}, \boldsymbol{x}) = \Phi_H(\boldsymbol{S}, \boldsymbol{x})$ for all $\boldsymbol{S}$ and $\boldsymbol{x}$.

The generalization of the construction described here to transform any SPGNN $\Psi$ into a SPGNN $\Phi$ that only uses order 1 polynomials by replacing every filter $H$ in $\Psi$ by the corresponding $\Phi_H$ is straightforward.

*Proof.* We consider a SPGNN $\Psi_H$ of the form $\Psi_H(\boldsymbol{S}, \boldsymbol{x}) = H(\boldsymbol{S})\boldsymbol{x}$ where $H(X) = \sum\limits_{i=0}^{d} h_i X^i$ is a polynomial of degree $d \geq 2$. (If $d \leq 1$, then the definition $\Phi_H = \Psi_H$ satisfies the property.) As shown in Figure 4, we define $\Phi_H$ from $\Psi_H$ as the $d$ layer SPGNN with two features per intermediate layer and filters:

- $H_{1,1}^{(0)} = h_d X + h_{d-1}$, $H_{1,2}^{(0)} = 1$

- For $1 \leq l \leq d-2$, $H_{1,1}^{(l)} = X$, $H_{1,2}^{(l)} = 0$, $H_{2,1}^{(l)} = h_{d-(l+1)}X$, $H_{2,2}^{(l)} = 1$.

- $H_{1,1}^{(d-1)} = X$, $H_{2,1}^{(d-1)} = h_0$.

The equality $\Phi_H(\boldsymbol{S}, \boldsymbol{x}) = H(\boldsymbol{S})\boldsymbol{x} = \Psi_H(\boldsymbol{S}, \boldsymbol{x})$ is simply a rewriting of the equality $h_d X^d + h_{d-1}X^{d-1} + \cdots + h_0 = X(\ldots X(X(h_d X + h_{d-1}) + h_{d-2}) + \cdots + h_1) + h_0$. $\qquad\square$

Then, since we have the equality $||\Psi_H(\boldsymbol{S}, \boldsymbol{x}) - \Psi_H(\hat{\boldsymbol{S}}, \boldsymbol{x})|| = ||\Phi_H(\boldsymbol{S}, \boldsymbol{x}) - \Phi_H(\hat{\boldsymbol{S}}, \boldsymbol{x})||$ by definition of $\Phi_H$, and since $\Phi_H$ now satisfies hypotheses of Theorem 2, we have the inequality:

$$||\Psi_H(\boldsymbol{S}, \boldsymbol{x}) - \Psi_H(\hat{\boldsymbol{S}}, \boldsymbol{x})|| \leq B_H ||S - \hat{S}||_{op}||\boldsymbol{x}||$$

where $B_H$ is the bound of Theorem 2 applied to $\Phi_H$.

To complete this analysis, we can compute explicitly $B_H$. We fix two GSOs $\boldsymbol{S}, \hat{\boldsymbol{S}}$ and we note $\Lambda := \max\limits_{\lambda \in \boldsymbol{S}}|\lambda|$ and $\hat{\Lambda} := \max\limits_{\lambda \in \hat{\boldsymbol{S}}}|\lambda|$ their largest eigenvalues. The proof of Theorem 4 yields the expression:

$$B_H = \sum_{\boldsymbol{f} \in Lay^{(d)}} \sum_{k=1}^{d} \left[ \Big( \prod_{l=d-k+1}^{d-1} ||H_{\boldsymbol{f}_l, \boldsymbol{f}_{l+1}}^{(l)}||_\infty \Big) \cdot A_{\boldsymbol{f}_{d-k}, \boldsymbol{f}_{d-k+1}}^{(d-k)} \cdot \Big( \prod_{l=0}^{d-k-1} ||\hat{H}_{\boldsymbol{f}_l, \boldsymbol{f}_{l+1}}^{(l)}||_\infty \Big) \right]$$

where $Lay^{(d)} = \{1\} \times \{1,2\}^{d-1} \times \{1\}$ and

- $||\hat{H}_{1,1}^{(0)}||_\infty = \max_{\lambda \in \sigma(\boldsymbol{S})} |h_d \lambda + h_{d-1}|$ and $A_{1,1}^{(0)} = |h_d|$,

- $||H_{1,2}^{(0)}||_\infty = 1$ and $A_{1,2}^{(0)} = 0$,

- for $l \in \{1, \dots d-1\}$, $||H_{1,1}^{(l)}||_\infty = \Lambda$, $||\hat{H}_{1,1}^{(l)}||_\infty = \hat{\Lambda}$, $||H_{1,2}^{(l)}||_\infty = ||\hat{H}_{1,2}^{(l)}||_\infty = 0$, $||H_{2,1}^{(l)}||_\infty = ||\hat{H}_{2,1}^{(l)}||_\infty = |h_{d-(l+1)}|$, $||H_{2,2}^{(l)}||_\infty = ||\hat{H}_{(2,2)}^{(l)}||_\infty = 1$ and
  $A_{1,1}^{(l)} = 1$, $A_{1,2}^{(l)} = A_{2,1}^{(l)} = A_{2,2}^{(l)} = 0$,

- $||H_{1,1}^{(d-1)}||_\infty = \Lambda$ and $A_{1,1}^{(d-1)} = 1$,

- $||H_{2,1}^{(d-1)}||_\infty = |h_0|$ and $A_{2,1}^{(d-1)} = 0$.

If we note $\lambda_d$ a value such that $|h_d \lambda_d + h_{d-1}| = \max_{\lambda \in \sigma(\boldsymbol{S})} |h_d \lambda + h_{d-1}|$, we can write:

$$B_H = |h_d \lambda_d + h_{d-1}| \Big( \sum_{l=1}^{d-1} \hat{\Lambda}^{l-1} * \Lambda^{d-(l+1)} \Big) + |h_d| \hat{\Lambda}^{d-1} + |h_{d-2}| \Big( \sum_{l=2}^{d-1} \hat{\Lambda}^{l-2} * \Lambda^{d-(l+1)} \Big) + \dots + |h_1|$$

.

Using the inequality $|h_d \lambda_d + h_{d-1}| \le |h_d| \Lambda + |h_{d-1}|$ and redefining $\Lambda = \max\{\Lambda, \hat{\Lambda}\}$ for better readability, we finally obtain: $B_H \le \sum_{i=1}^{d} i |h_i| \Lambda^{i-1}$. Note that this bound is a bound to the derivative of $H$, which is highly natural for an inequality on $||H(\boldsymbol{S}) - H(\hat{\boldsymbol{S}})||_{op}$. To match this intuition with the notations, we now write $\Delta(H) = \sum_{i=1}^{d} i |h_i| \Lambda^{i-1}$.

So far, we have shown that $||H(\boldsymbol{S})\boldsymbol{x} - H(\hat{\boldsymbol{S}})\boldsymbol{x}|| \le B_H ||\boldsymbol{S} - \hat{\boldsymbol{S}}||_{op}\boldsymbol{x}$, in particular $||H(\boldsymbol{S}) - H(\hat{\boldsymbol{S}})||_{op} \le B_H ||\boldsymbol{S} - \hat{\boldsymbol{S}}||_{op}$.

Now remind the remark made at the end of the proof of Theorem 2: the whole proof on relies on bounding $||H(\boldsymbol{S})||_{op}$ and $||H(\boldsymbol{S}) - H(\hat{\boldsymbol{S}})||_{op}$ for a general filter $H$. The definition of $||H||_\infty := \max_{\lambda \in \sigma(\boldsymbol{S})} |H(\lambda)|$ still satisfies $||H(\boldsymbol{S})||_{op} = ||H||_\infty$, and the property we have just shown is a bound to $||H(\boldsymbol{S}) - H(\hat{\boldsymbol{S}})||_{op}$. Therefore, Theorem 2 applies and states, for a SPGNN $\Psi$ that uses filters that are polynomials of any order, with the same notations as in the original statement:

$$||\Psi(\boldsymbol{S}, \boldsymbol{x}) - \Psi(\hat{\boldsymbol{S}}, \boldsymbol{x})|| \le B^{(L)} ||\boldsymbol{S} - \hat{\boldsymbol{S}}||_{op} ||\boldsymbol{x}||$$

where $B^{(L)} = \sum_{\boldsymbol{f} \in Lay^{(L)}} \sum_{k=1}^{L} \left[ \Big( \prod_{l=L-k+1}^{L-1} ||H_{\boldsymbol{f}_l, \boldsymbol{f}_{l+1}}^{(l)}||_\infty \Big) \cdot B_{H_{\boldsymbol{f}_{L-k}, \boldsymbol{f}_{L-k+1}}^{(L-k)}} \cdot \Big( \prod_{l=0}^{L-k-1} ||\hat{H}_{\boldsymbol{f}_l, \boldsymbol{f}_{l+1}}^{(l)}||_\infty \Big) \right]$.

This can be written in the simpler form:

$$||\Psi(\boldsymbol{S}, \boldsymbol{x}) - \Psi(\hat{\boldsymbol{S}}, \boldsymbol{x})|| \le L\Delta(H)_\infty (F||H||_\infty)^{L-1} ||\boldsymbol{S} - \hat{\boldsymbol{S}}||_{op} ||\boldsymbol{x}||$$

where $\Delta(H)_\infty = \max_{H \in \Psi} \Delta(H)$.

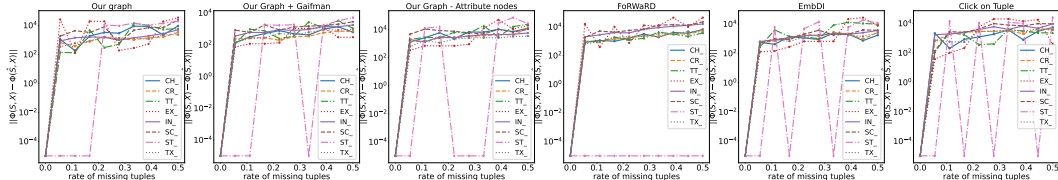

Figure 5: Average distances between the tuple embeddings of each relation of clean database and their counterpart from the perturbed database. The perturbed database has been generated from the clean one where tuples have been removed at a rate from 0 up to 0.5. The neural network is composed of 12 layers; a minor perturbation in the database can impact the embedding of all elements of the database.

### A.4.2 HYPOTHESIS ON THE SYMMETRY OF THE GSOs

Once again, we use the remark made at the end of the proof to Theorem 2: if we can provide alternative bounds for $||H(\boldsymbol{S})||_{op}$ and $||H(\boldsymbol{S}) - H(\hat{\boldsymbol{S}})||_{op}$, we have a new version of the result. If we assume that the filters are order 1 polynomials, then $||H(\boldsymbol{S}) - H(\hat{\boldsymbol{S}})||_{op} \leq A||\boldsymbol{S} - \hat{\boldsymbol{S}}||_{op}$ and therefore Theorem 2 yields, for a SPGNN $\Phi$:

$$||\Phi(\boldsymbol{S}, \boldsymbol{x}) - \Phi(\hat{\boldsymbol{S}}, \boldsymbol{x})|| \leq B^{(L)}||\boldsymbol{S} - \hat{\boldsymbol{S}}||_{op}||\boldsymbol{x}||$$

where $B^{(L)} = \sum_{\boldsymbol{f} \in Lay^{(L)}} \sum_{k=1}^{L} \left[ \left( \prod_{l=L-k+1}^{L-1} ||H^{(l)}_{\boldsymbol{f}_l, \boldsymbol{f}_{l+1}}(\boldsymbol{S})||_{op} \cdot A^{(L-k)}_{\boldsymbol{f}_{L-k}, \boldsymbol{f}_{L-k+1}} \cdot \left( \prod_{l=0}^{L-k-1} ||H^{(l)}_{\boldsymbol{f}_l, \boldsymbol{f}_{l+1}}(\hat{\boldsymbol{S}})||_{op} \right) \right]$.

This can be written in the simpler form:

$$||\Phi(\boldsymbol{S}, \boldsymbol{x}) - \Phi(\hat{\boldsymbol{S}}, \boldsymbol{x})|| \leq LA(F||H||_{op,\infty})^{L-1}||\boldsymbol{S} - \hat{\boldsymbol{S}}||_{op}||\boldsymbol{x}||$$

where $||H||_{op,\infty} = \max_{H \in \Phi} ||H(\boldsymbol{S})||_{op}, ||H(\hat{\boldsymbol{S}})||_{op}$.

Finally, we can combine both of the generalizations and they yield, for a SPGNN $\Psi$ with filters of any orders:

$$||\Psi(\boldsymbol{S}, \boldsymbol{x}) - \Psi(\hat{\boldsymbol{S}}, \boldsymbol{x})|| \leq B^{(L)}||\boldsymbol{S} - \hat{\boldsymbol{S}}||_{op}||\boldsymbol{x}||$$

where $B^{(L)} = \sum_{\boldsymbol{f} \in Lay^{(L)}} \sum_{k=1}^{L} \left[ \left( \prod_{l=L-k+1}^{L-1} ||H^{(l)}_{\boldsymbol{f}_l, \boldsymbol{f}_{l+1}}(\boldsymbol{S})||_{op} \cdot B_{H^{(L-k)}_{\boldsymbol{f}_{L-k}, \boldsymbol{f}_{L-k+1}}} \cdot \left( \prod_{l=0}^{L-k-1} ||H^{(l)}_{\boldsymbol{f}_l, \boldsymbol{f}_{l+1}}(\hat{\boldsymbol{S}})||_{op} \right) \right]$

and, in that context, for a filter $H$, $B_H = ||h_d S + h_{d-1}||_{op}(\sum_{l=1}^{d-1} ||\hat{\boldsymbol{S}}||_{op}^{l-1} * ||\boldsymbol{S}||_{op}^{d-(l+1)}) + |h_d|||\hat{\boldsymbol{S}}||_{op}^{d-1} + |h_{d-2}|(\sum_{l=2}^{d-1} ||\hat{\boldsymbol{S}}||_{op}^{l-2} * ||\boldsymbol{S}||_{op}^{d-(l+1)}) + \cdots + |h_1|$.

This can be written in the simpler form:

$$||\Psi(\boldsymbol{S}, \boldsymbol{x}) - \Psi(\hat{\boldsymbol{S}}, \boldsymbol{x})|| \leq L\tilde{\Delta}(H)_{\infty}(F||H||_{op,\infty})^{L-1}||\boldsymbol{S} - \hat{\boldsymbol{S}}||_{op}||\boldsymbol{x}||$$

where $\tilde{\Delta}(H)_{\infty} = \max_{H \in \Psi, \boldsymbol{M} \in \{\boldsymbol{S}, \hat{\boldsymbol{S}}\}} \sum_{i=1}^{d} i|h_i| \cdot ||\boldsymbol{M}||_{op}^{i-1}$.

## B  OTHER EXPERIMENTAL DETAILS

Table 2 shows the detail of the TPC-E tables used to generate the database graphs.

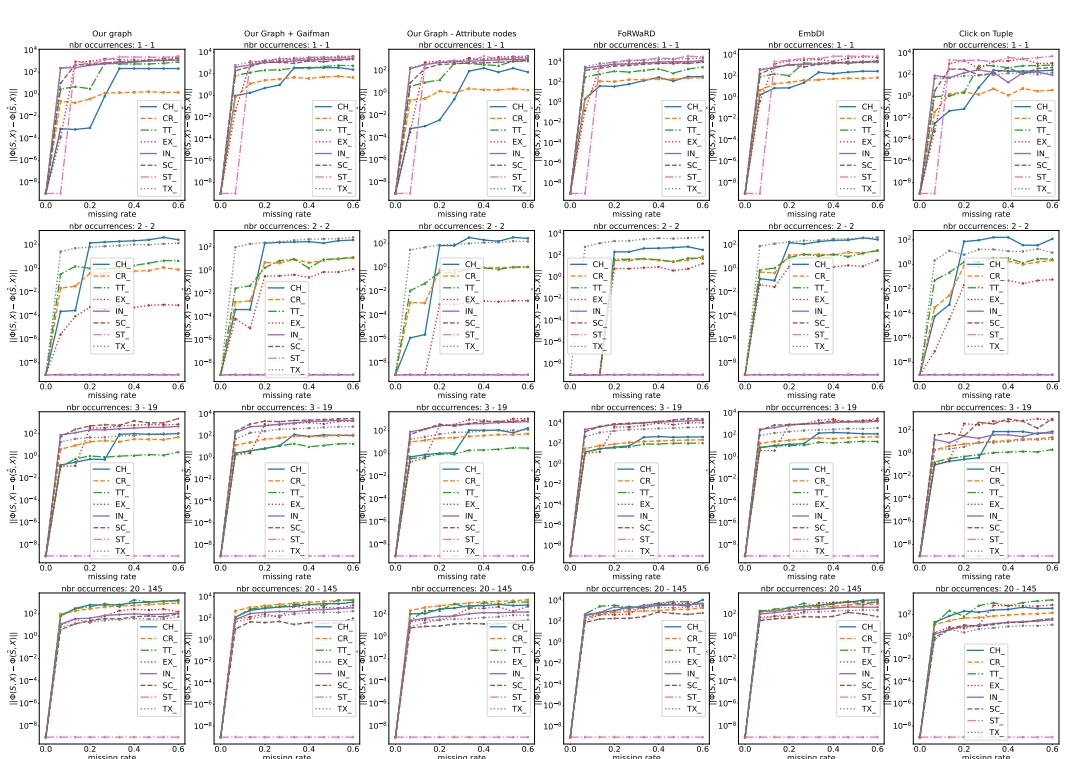

Figure 6: Average distances between the tuple embeddings of each relation of clean database and their counterpart from the perturbed database. The perturbed database has been generated from the clean one where values have been removed at a rate from 0 up to 0.5. The neural network is composed of 12 layers; a minor perturbation in the database can impact the embedding of all elements of the database.

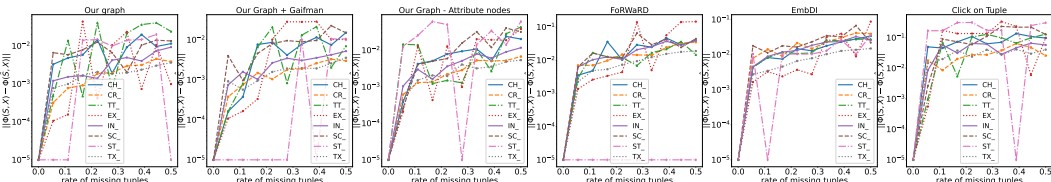

Figure 7: Average distances between the tuple embeddings of each relation of clean database and their counterpart from the perturbed database. The perturbed database has been generated from the clean one where tuples have been removed at a rate from 0 up to 0.5. The neural network is composed of 3 layers; a minor perturbation in the database is less likely to impact the embedding of all elements of the database.

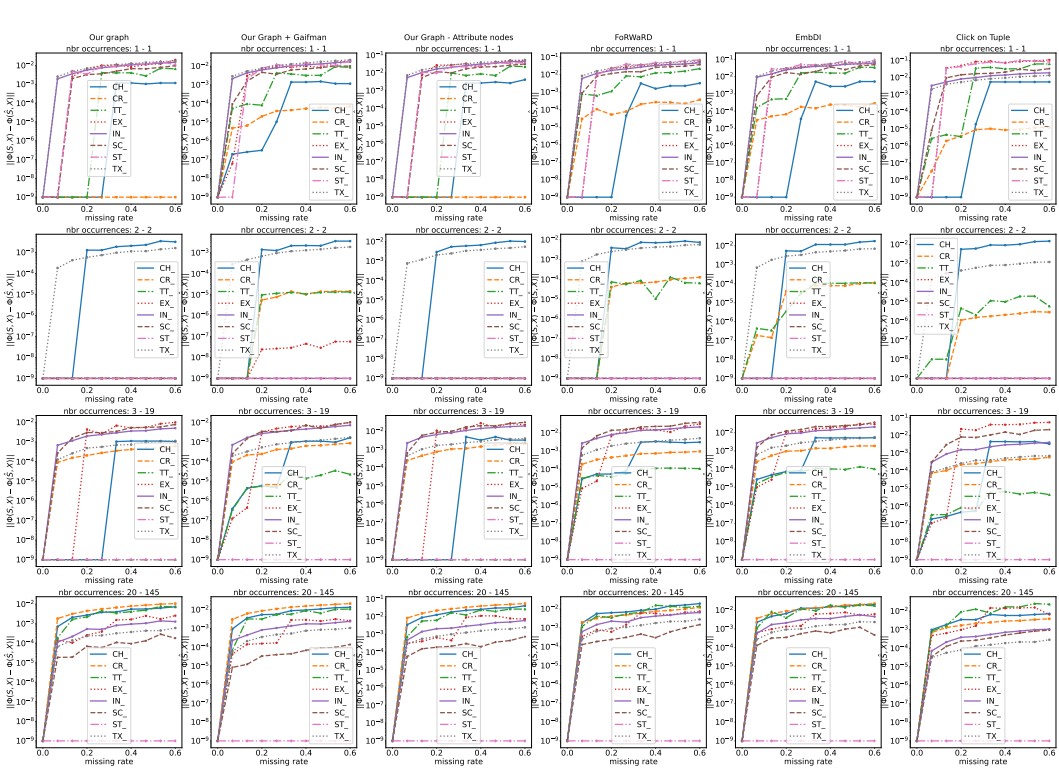

Figure 8: Average distances between the tuple embeddings of each relation of clean database and their counterpart from the perturbed database. The perturbed database has been generated from the clean one where values have been removed at a rate from 0 up to 0.5. The neural network is composed of 3 layers; a minor perturbation in the database is less likely to impact the embedding of all elements of the database.

Table 2: Detail of TPC-E tables used.

| Table | ISO Prefix | Tuples | Attributes |
|---|---|---|---|
| Security | S_ | 154 | 16 |
| Broker | B_ | 49 | 5 |
| TaxRate | TX_ | 169 | 3 |
| Company | CO_ | 112 | 9 |
| Industry | IN_ | 101 | 3 |
| Sector | SC_ | 11 | 2 |
| Customer | C_ | 145 | 24 |
| CommissionRate | CR_ | 134 | 6 |
| ZipCode | ZC_ | 134 | 3 |
| Financial | FI_ | 142 | 14 |
| CustomerTaxrate | CX_ | 180 | 2 |
| WatchList | WL_ | 117 | 2 |
| Charge | CH_ | 14 | 3 |
| Exchange | EX_ | 3 | 7 |
| HoldingSummary | HS_ | 126 | 3 |
| WatchItem | WI_ | 204 | 2 |
| TradeType | TT_ | 4 | 4 |
| LastTrade | LT_ | 119 | 5 |
| AccountPermission | AP_ | 135 | 5 |
| NewsXRef | NX_ | 164 | 2 |
| Address | AD_ | 107 | 5 |
| CustomerAccount | CA_ | 118 | 6 |
| CompanyCompetitor | CP_ | 199 | 3 |
| StatusType | ST_ | 4 | 2 |

**Other comparison with GNN variants** We also compare the degree one SPGNNs on other datasets, CitySeer Yang et al. (2016) and PubMed Yang et al. (2016) along with the Cora dataset. CitySeer is a graph composed of 3,327 nodes and 9,104 edges. Each node belongs to one of the six classes which compose the corpus and has an initial feature mappings of dimension 3,703. PubMed is another graph dataset composed of 19,717 node distributed among three different classes. These nodes are connected by 88,648 edges and have an initial feature map of dimension 500. The experimental conditions remain the same as the ones presented in Section 6 and the results are presented in Table 3 and reinforce the results presented in the main text. The table presents the accuracy scores performed by the different GNNs on graphs that keep 100%, 90%, 70%, 50%, 30% of their original nodes. The annotation (1) identify the experiments where the 2-layer MLP is retrained for each "dirty" graph while (2) denote the experiments where none of the GNNs nor the MLP are retrained.

**Other databases** We have used the following other datasets from the relational database domain to corroborate the results in Section 6:

- **genes** Cheng et al. (2002) This dataset has been presented for the 2001 KDD competition, containing genomics and drug design related data.

- **hepatitis** Neville et al. (2003) a database which presents biological conditions of patients having contracted either hepatitis A or B.

- **mondial** May (1999) a dataset which shows whether a country is is predominantly Christian or not.

- **mutagenesis** Lodhi & Muggleton (2005) a dataset composed of data concerning mutagenicity of molecules on Salmonella typhimurium.

- **world** Toenshoff et al. (2023) a dataset which contains data on countries and their cities.

The smaller versions of these databases, used in this work, are presented in Table 4 and their associated graphs in Table 5.

Table 3: SPGNN vs. state of the art

|  | GCN (1) | GAT (1) | SPGNN (1) | GCN (2) | GAT (2) | SPGNN (2) |
|---|---|---|---|---|---|---|
| **Cora (100%)** | $0.86 \pm 0.02$ | $\mathbf{0.87 \pm 0.01}$ | $0.85 \pm 0.05$ | $0.86 \pm 0.02$ | $\mathbf{0.87 \pm 0.01}$ | $0.85 \pm 0.05$ |
| **Cora (90%)** | $0.74 \pm 0.04$ | $0.76 \pm 0.018$ | $\mathbf{0.81 \pm 0.02}$ | $0.65 \pm 0.06$ | $0.67 \pm 0.06$ | $0.65 \pm 0.07$ |
| **Cora (70%)** | $0.71 \pm 0.03$ | $0.73 \pm 0.01$ | $\mathbf{0.80 \pm 0.02}$ | $0.62 \pm 0.04$ | $0.60 \pm 0.02$ | $0.58 \pm 0.04$ |
| **Cora (50%)** | $0.62 \pm 0.05$ | $0.62 \pm 0.05$ | $\mathbf{0.77 \pm 0.02}$ | $0.44 \pm 0.09$ | $0.44 \pm 0.07$ | $0.42 \pm 0.06$ |
| **Cora (30%)** | $0.49 \pm 0.07$ | $0.52 \pm 0.05$ | $\mathbf{0.71 \pm 0.04}$ | $0.24 \pm 0.07$ | $0.27 \pm 0.04$ | $0.28 \pm 0.10$ |
| **PubMed (100%)** | $0.84 \pm 0.01$ | $0.84 \pm 0.01$ | $\mathbf{0.86 \pm 0.01}$ | $0.84 \pm 0.01$ | $0.84 \pm 0.01$ | $\mathbf{0.86 \pm 0.01}$ |
| **PubMed (90%)** | $0.79 \pm 0.04$ | $0.78 \pm 0.05$ | $\mathbf{0.84 \pm 0.01}$ | $0.78 \pm 0.05$ | $0.79 \pm 0.03$ | $0.76 \pm 0.07$ |
| **PubMed (70%)** | $0.77 \pm 0.08$ | $0.76 \pm 0.02$ | $\mathbf{0.83 \pm 0.01}$ | $0.75 \pm 0.03$ | $0.74 \pm 0.02$ | $0.73 \pm 0.03$ |
| **PubMed (50%)** | $0.68 \pm 0.08$ | $0.67 \pm 0.08$ | $\mathbf{81 \pm 0.01}$ | $0.61 \pm 0.12$ | $0.61 \pm 0.12$ | $0.58 \pm 0.13$ |
| **PubMed (30%)** | $0.70 \pm 0.02$ | $0.68 \pm 0.02$ | $\mathbf{0.82 \pm 0.01}$ | $0.65 \pm 0.01$ | $0.65 \pm 0.02$ | $0.64 \pm 0.02$ |
| **CiteSeer (100%)** | $\mathbf{0.75 \pm 0.02}$ | $0.75 \pm 0.02$ | $0.71 \pm 0.04$ | $\mathbf{0.75 \pm 0.02}$ | $\mathbf{0.75 \pm 0.02}$ | $0.71 \pm 0.04$ |
| **CiteSeer (90%)** | $0.64 \pm 0.02$ | $0.61 \pm 0.04$ | $\mathbf{0.68 \pm 0.02}$ | $0.32 \pm 0.12$ | $0.32 \pm 0.12$ | $0.29 \pm 0.10$ |
| **CiteSeer (70%)** | $0.60 \pm 0.03$ | $0.60 \pm 0.02$ | $\mathbf{0.66 \pm 0.01}$ | $0.36 \pm 0.15$ | $0.36 \pm 0.15$ | $0.34 \pm 0.18$ |
| **CiteSeer (50%)** | $\mathbf{0.61 \pm 0.02}$ | $0.58 \pm 0.05$ | $0.65 \pm 0.05$ | $0.23 \pm 0.08$ | $0.25 \pm 0.09$ | $0.23 \pm 0.08$ |
| **CiteSeer (30%)** | $0.56 \pm 0.03$ | $0.53 \pm 0.04$ | $\mathbf{0.62 \pm 0.02}$ | $0.24 \pm 0.10$ | $0.23 \pm 0.10$ | $0.23 \pm 0.11$ |

Table 4: Properties of extra databases used.

| Database | Relations | Tuples | Attributes |
|---|---|---|---|
| genes | 3 | 300 | 15 |
| mutagenesis | 3 | 300 | 14 |
| mondial | 34 | 3170 | 136 |
| hepatitis | 7 | 632 | 26 |
| world | 3 | 300 | 24 |

Figures 9–14 show further results on the stability of the database embeddings for the value removal perturbation. These results corroborate the ones presented in Section 6 for TPC-E.

**Oversmoothing (Rusch et al. (2023))**  To ensure that our stability results are not simply a byproduct of oversmoothing, we compare here the stability results on the Hepatitis dataset, obtained with a 5-layer GNN (Fig. 10) with those obtained with a 2-layer GNN (Fig. 11)(mitigating oversmoothing). As we can observe similar trends, but at different scales, we can conclude that the initial experiments (Fig. 1-3) using GNNs with 3 to 5 layers are not significantly impacted by oversmoothing.

**Chaining tuple deletions and additions**  We conduct experiments where we sequentially add or delete tuples in a database. The rationale for this experiment is to confirm that adding and removing values are symmetrical operations w.r.t. stability. The results are present in Fig. 15-17. Note

Table 5: Sizes of graphs generated from databases in Table 4.

|  |  | FoRWaRD | EmBDi | 5-DB | 5-DB Gaif | 5-DB-Att | Tuple+Clique |
|---|---|---|---|---|---|---|---|
| genes | nodes | 7,902 | 8,041 | 21,349 | 21,210 | 21,210 | 7,902 |
|  | edges | 11,698 | 20,048 | 41,843 | 46,058 | 28,569 | 161,448 |
| mondial | nodes | 10,089 | 10,225 | 26,377 | 26,241 | 26,241 | 10,089 |
|  | edges | 15,131 | 24,571 | 50,748 | 96,246 | 34,305 | 205,168 |
| mutagenesis | nodes | 706 | 720 | 2,123 | 2,109 | 2,109 | 706 |
|  | edges | 1,351 | 1,906 | 4,500 | 4,830 | 3,100 | 16,201 |
| world | nodes | 1,792 | 1,816 | 4,219 | 4,195 | 4,195 | 1,792 |
|  | edges | 2,266 | 3,923 | 7,698 | 15,340 | 5,098 | 17,116 |
| hepatitis | nodes | 820 | 846 | 3,249 | 3,223 | 3,223 | 820 |
|  | edges | 1,782 | 2,597 | 7,820 | 6,211 | 5,424 | 31,978 |

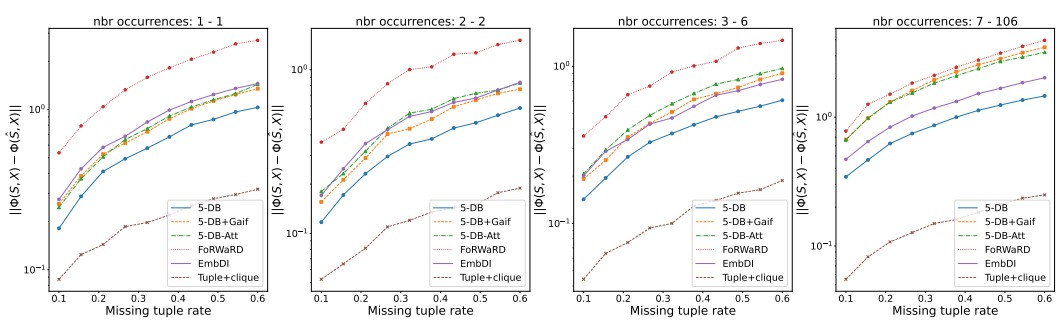

Figure 9: Stability to missing values, by removing up to $90\%$ of the values of the database. We show result for removal of values of various frequency bins. This experiment has been performed on the Genes database.

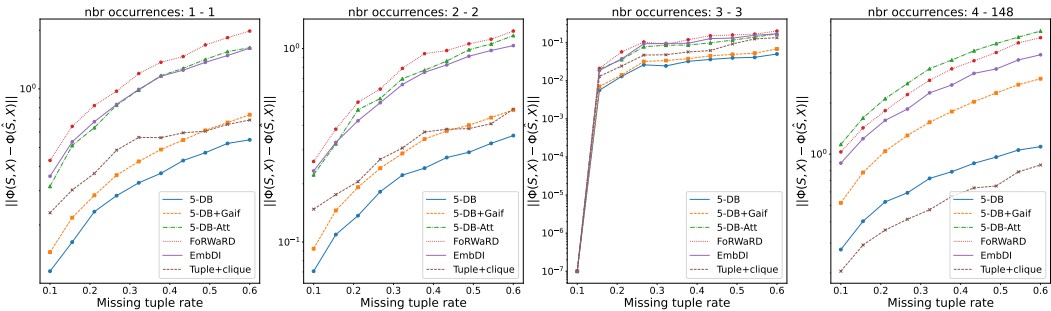

Figure 10: Stability to missing values, by removing up to $90\%$ of the values of the database. We show result for removal of values of various frequency bins. This experiment has been performed on the Mutagenesis database.

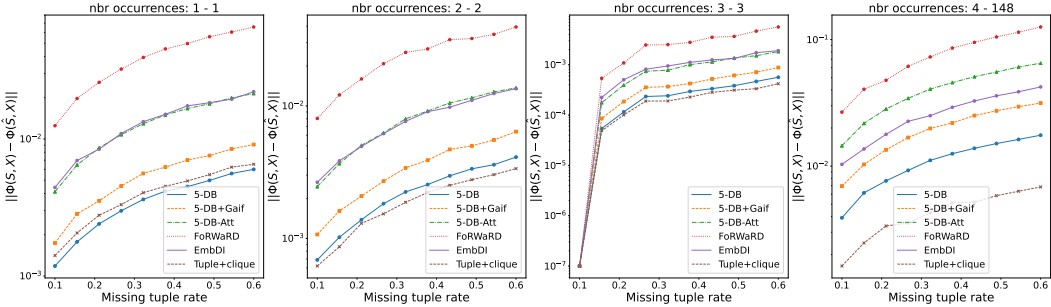

Figure 11: Stability to missing values, by removing up to $90\%$ of the values of the database. We show result for removal of values of various frequency bins. This experiment has been performed on the Mutagenesis database using a *2-layer* SPGNN.

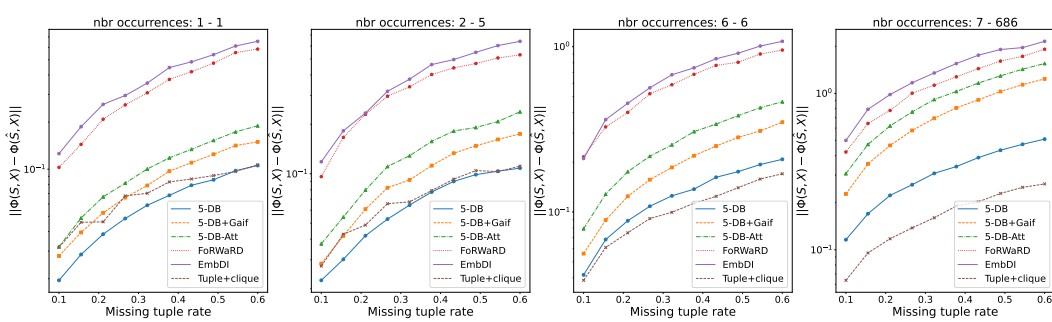

Figure 12: Stability to missing values, by removing up to $90\%$ of the values of the database. We show result for removal of values of various frequency bins. This experiment has been performed on the Hepatitis database.

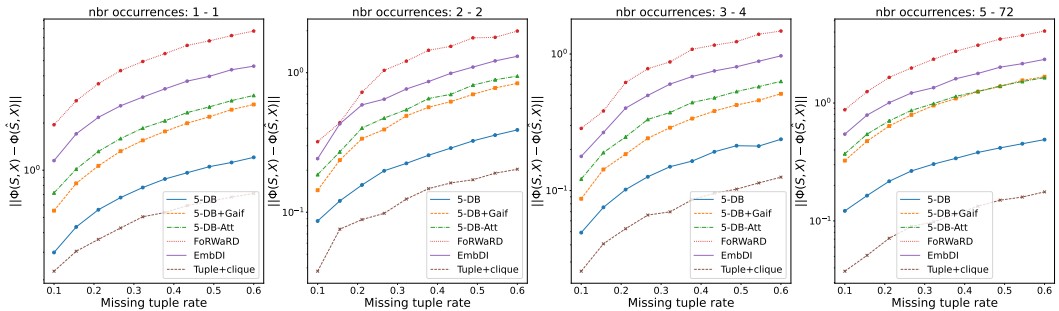

Figure 13: Stability to missing values, by removing up to $90\%$ of the values of the database. We show result for removal of values of various frequency bins. This experiment has been performed on the World database.

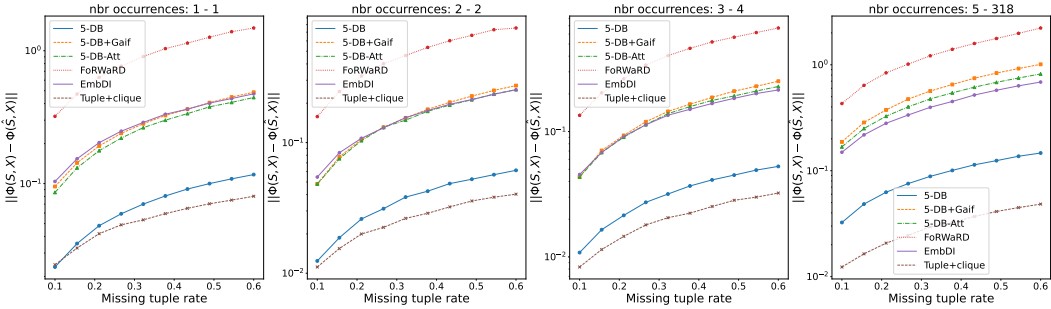

Figure 14: Stability to missing values, by removing up to $90\%$ of the values of the database. We show result for removal of values of various frequency bins. This experiment has been performed on the Mondial database.

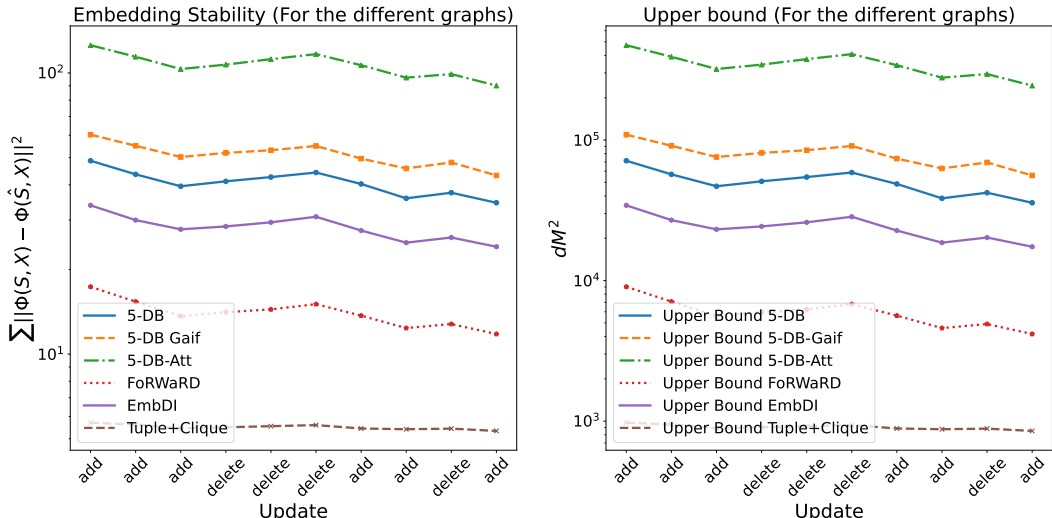

Figure 15: Experiment performed over the Hepatitis database Neville et al. (2003). The plot on the left explores the distance between (a) the embeddings generated from a truncated database $\mathbb{D}_1$ in which, at each step, some tuples are either added or deleted, and (b) the embeddings generated using the entire database $\mathbb{D}_2$. To plot on the right presents the upper bounds defined in Corollary 2.1; $\mathbb{D}_1$ has been preprocessed s.t., at the first iteration, 50% of its tuples are missing. Each addition increments the database by 20% of the left-out tuples. For each removal operation, 10% of the total number of tuples are removed.

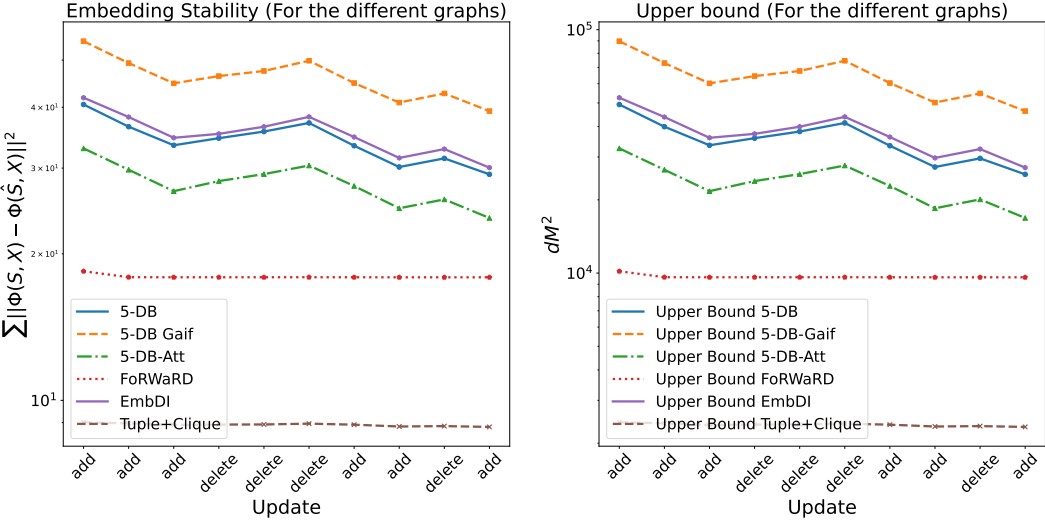

Figure 16: Experiment performed over the Mutagenesis database Lodhi & Muggleton (2005). The plot on the left explores the distance between (a) the embeddings generated from a truncated database $\mathbb{D}_1$ in which, at each step, some tuples are either added or deleted, and (b) the embeddings generated using the entire database $\mathbb{D}_2$. To plot on the right presents the upper bounds defined in Corollary 2.1; $\mathbb{D}_1$ has been preprocessed s.t., at the first iteration, 50% of its tuples are missing. Each addition increments the database by 20% of the left-out tuples. For each removal operation, 10% of the total number of tuples are removed.

that the results presented in the main experimental section focus on the worst-case scenario, by examining the distance between two node embeddings of the two graphs. By applying Corollary 2.1 and summing over all embeddings, one can observe that the practical and theoretical results are much closer.

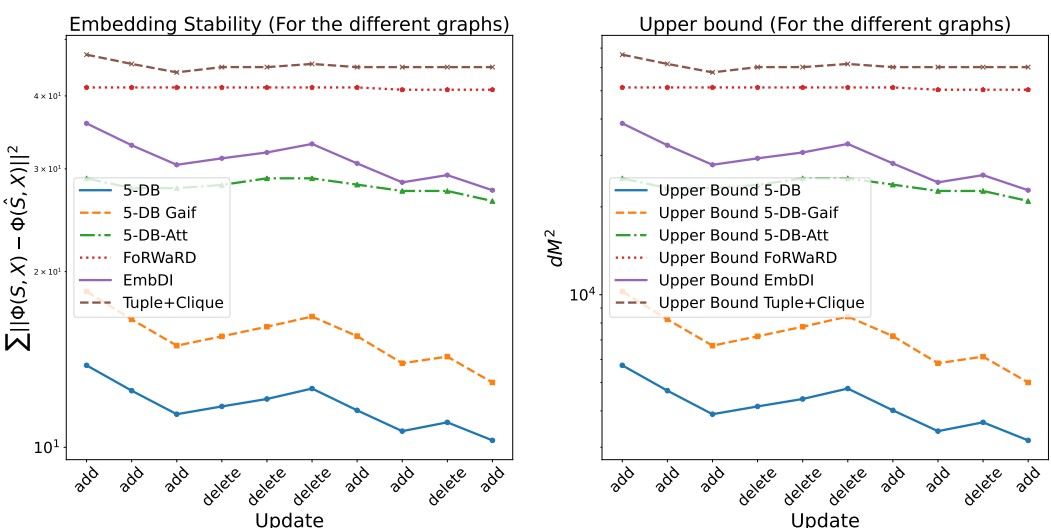

Figure 17: Experiment performed over the World database Toenshoff et al. (2023). The plot on the left explores the distance between (a) the embeddings generated from a truncated database $\mathbb{D}_1$ in which, at each step, some tuples are either added or deleted, and (b) the embeddings generated using the entire database $\mathbb{D}_2$. To plot on the right presents the upper bounds defined in Corollary 2.1; $\mathbb{D}_1$ has been preprocessed s.t., at the first iteration, 50% of its tuples are missing. Each addition increments the database by 20% of the left-out tuples. For each removal operation, 10% of the total number of tuples are removed.