# OpenReview forum: "Stable GNN Embeddings for Relational Data"
_ICLR.cc/2025/Conference — Submitted to ICLR 2025_

### Official Review · Reviewer_Ffdv · 2024-11-03

**Soundness:** 3
**Presentation:** 2
**Contribution:** 3
**Rating:** 6
**Confidence:** 3

**Summary:**

The paper studies the stability, i.e., minimal variation under perturbation, of node embeddings for relational databases under common database operation (e.g., value or node deletions). To this end, the paper builds on previous work from Lama et al. using the Signal Processing Graph Neural Network (SPGNN) model, and derives extended results. In particular, the paper proves a new stability bound that crucially does not depend on the number of nodes in the input graph. The paper then derives results based on individual node embeddings, first showing how the general bound in Theorem 2 leads to an asymptotic stability bound on at least half the nodes, subject to SPGNN instantiations where graph shift operations (GSOs) have eigenvalues that do not increase with the number of nodes $N$. The paper then provides an individual node-level bound based on the notion of $k-$hop equality, showing that nodes with more similar subgraphs across perturbations tend to remain more closely embedded. The paper also discusses the operations can be studied within this framework (tuple and value removal), and how these can be reflected in the GSO comparison.

Empirically, the paper conducts experiments on Cora, showing that a 1st-order SPGNN (as is specified in the Theorem 2 result) better maintains its accuracy (only when re-trained) than standard GNN baselines on subsets of Cora following node removals, and shows that node degrees have a minor, but not problematic effect of node embedding stability.  Finally, the paper discusses and empirically compares different relational database graph encoding techniques on a subset of the TPC-E dataset, including their own proposed 5-DB encoding, showing that their encoding is more resilient for tuple removal, but that denser encodings (at the value) like Tuple+Clique are more resilient in the value removal context.

**Strengths:**

- The theoretical results are interesting, and appear sound (although I must admit that I did not check these in detail).
- The empirical setup comparing different encodings on TPC-E is informative, and the tasks being studied are a useful baseline for this area of research.
- The discussion of the results, and the overall flow of the paper, is easy to follow.

**Weaknesses:**

- The empirical results on relational databases need scaffolding with a signal from a machine learning task, e.g., predicting missing edges, tuple classification, etc., to place the observed stability into context. While stability in itself is valuable, it's important that this pairs with robust model performance. This is somewhat covered in the Cora experiment through accuracy, but not done in the TPC-E setting. I strongly recommend the authors add a task dimension to complement their existing study. This would also add more information about the merits of the corresponding relational database graph encoding schemes.
- I find that the experimental analysis could be better served with more detailed explanations of how the bounds apply on a smaller synthetic task, i.e., a case study. As it stands, the TPC-E experiment demonstrates a substantial difference (roughly 6 orders of magnitude) between observed and theoretical stability, with not much discussion accompanying this observation. It just leaves it harder to make the connection between the experiments and the theoretical contribution. The authors, to their credit, acknowledge that their node-level bounds are less intuitive, and so adding some experiments that are more fundamental and small-scale, i.e., toy experiments, to illustrate the bounds in action would be a very insightful addition to the work.

**Questions:**

Please address weaknesses mentioned above.

---

> ### Author Response · Authors · 2024-11-21
> **rebuttal on review Ffdv**
>
> > The empirical results on relational databases need scaffolding with a signal from a machine learning task, e.g., predicting missing edges,  tuple classification, etc., to place the observed stability into context. While stability in itself is valuable, it's important that this pairs with robust model performance. This is somewhat covered in the Cora experiment through accuracy, but not done in the TPC-E setting. I strongly recommend the authors add a task dimension to complement their existing study. This would also add more information about the merits of the corresponding relational database graph encoding schemes.
>
> Thank you for the suggestion. This paper primarily focuses on database-to-graph constructions and the stability of resulting embeddings in the face of incomplete or perturbed data. While we acknowledge the importance of other downstream tasks, such as tuple classification, missing value imputation, or entity resolution, a comprehensive evaluation of these tasks was beyond the scope of this work.
>
> > I find that the experimental analysis could be better served with more detailed explanations of how the bounds apply on a smaller synthetic task, i.e., a case study. As it stands, the TPC-E experiment demonstrates a substantial difference (roughly 6 orders of magnitude) between observed and theoretical stability, with not much discussion accompanying this observation. It just leaves it harder to make the connection between the experiments and the theoretical contribution. The authors, to their credit, acknowledge that their node-level bounds are less intuitive, and so adding some experiments that are more fundamental and small-scale, i.e., toy experiments, to illustrate the bounds in action would be a very insightful addition to the work.
>
> Thank you for the insightful suggestion. Theoretical bounds are often loose in practice, as they typically consider worst-case scenarios. This is evident in Corollary 2.1, where the average-case bound is significantly tighter. Our new experiments on sequential data modifications (additions and removals), in Appendix, Figures 15-17, further corroborate this finding, revealing stability measures that align more closely with this theoretical bound (within one order of magnitude).

---

> ### Comment · Reviewer_Ffdv · 2024-11-26
> **Reviewer Response**
>
> I thank the authors for their rebuttal. I understand your points on scope and on my theoretical suggestion. All in all, I will maintain my rating.

---

### Official Review · Reviewer_V2r2 · 2024-11-03

**Soundness:** 2
**Presentation:** 1
**Contribution:** 2
**Rating:** 3
**Confidence:** 3

**Summary:**

This paper focuses on analysing the stability of signal processing GNNs (SPGNNs) by establishing limits for both node-level and graph-level embeddings. It also studies how to effectively transform relational databases into graphs towards better graph representation learning on relational databases.

**Strengths:**

- The paper provides a comprehensive and thorough study on the stability of Signal Processing GNNs by showing a more refined bound that no longer depends on the number of nodes, which is interesting.

- The graph construction for relational databases is under-explored, and this work presents a novel approach that are empirically outperforming existing transformations methods.

- Empirically, the authors show that SPGNNs are stable under shifts in the underlying databases.

**Weaknesses:**

- The topic and the focus of the paper is two-folded which are loosely related. It would be ideal if the paper can focus on one particular aspect, such as the stability of GNNs only (with potentially further results). In the current presentation, the respective parts of the paper read like two separate papers.

- While the experiments provide valuable insights, the scope of the chosen datasets is rather limited. It would be better if the authors could validate the observation on additional baselines, especially the heterophilic datasets [1]? Relatedly, it would be interesting to know whether the proposed graph transformation can be applies to new relational database benchmark [2]?

- There are quite a few presentation issues in the mathematical notation. For example:
  -The definition of operator norm (line 247) should come before its use (Theorem 1);
  - What are $v_i$ and $\hat{v}_i$ in Theorem 2, equation 3? Two corresponding nodes from the original and edited graphs, respectively? Two different ones?

- Minor issues: Figures 5-7 in the appendix are hardly readable. Also isn't there a $\delta$ missing in the statement of Theorem 1?

**Questions:**

- How would the results be affected when applied on standard heterophilic graph datasets in [1], both theoretically and empirically?
- Are the results directly applicable to link-level prediction? If not, can the authors comment on whether and how it could be adapted?
- Can you please elaborate on why WLOG Theorem 2 can be directly applied to higher-degree polynomial as the filter? Specifically, how do you deal with the non-linearity? (Remark 4)
- In Theorem 2, it is stated that the shift operators must be symmetric: is this not an unrealistic assumption, since  the edges are typically directed in relational databases (making the shift operators not symmetric)?
- Is transformation from relational database to graph really needed? Why should we model relational database as graph given that there is already a lot of models dedicated for reasoning over tabular data?
- Figure 2 shows that the theoretical upper-bound is of up to 8 order of magnitude larger than the empirical average distance. While the empirical results are quite interesting, are the bounds of any use, given that they are so loose?
- Are there any similar results (theoretical or empirical) on other models, for instance Graph Transformers [3] or even GATs [4]?
- On the empirical side, how can we distinguish stability effect from over-smoothing [5], especially when operating on such large graphs? (It might be better to conduct experiments with very few layers).

[1] Platonov, O., Kuznedelev, D., Diskin, M., Babenko, A., and Prokhorenkova, L. A critical look at the evaluation of
GNNs under heterophily: Are we really making progress? ICLR, 2023.

[2] Robinson, Joshua, et al. Relbench: A benchmark for deep learning on relational databases. arXiv preprint arXiv:2407.20060, 2024.

[3] Min, E., Chen, R., Bian, Y., Xu, T., Zhao, K., Huang, W., Zhao, P., Huang, J., Ananiadou, S., Rong, Y. Transformer for Graphs: An Overview from Architecture Perspective, arXiv preprint arXiv:2205.08455, 2022.

[4] Veličković, P., Cucurull, G., Casanova, A., Romero, A., Liò, P., Bengio, Y. Graph Attention Networks, ICLR 2018.

[5] Rusch, TK, Bronstein, MM, Mishra, S. A Survey on Oversmoothing in Graph Neural Networks, arXiv preprint arxiv:2303.10993, 2023.

---

> ### Author Response · Authors · 2024-11-21
> **rebuttal on review V2r2**
>
> >...paper 2-folded..loosely related..parts read like separate papers
>
> Our objective was to establish the applicability of embeddings derived from relational databases for downstream tasks **and** for direct latent space querying. Achieving this 2-fold objective is a substantial step forward. Stability, which guarantees the robustness of embeddings against database updates, is therefore fundamental to the construction of robust vector databases. **We will strengthen the paper's coherence by explicitly highlighting the crucial link between these objectives.** We have further emphasized this relationship in Sec 5 of the revised rebuttal.
>
> >limited scope of datasets..additional baselines, heterophilic datasets[1]..the new relational database benchmark[2]
>
> Our work centered on exploring methods for generating stable graph-based embeddings for relational databases. We believe the datasets we employed are representative enough to demonstrate that db-to-graph transformations can yield stable embeddings. It's important to note that TPC-E is a widely recognized and comprehensive benchmark for relational databases; we leveraged it to build our datasets.
>
> >Results affected if applied on standard heterophilic graph datasets[1]
>
> They would not be affected, our theoretical results remain valid for any graph, when used with SPGNNs
>
> >Results applicable to link prediction?
>
> Indeed, our stability results show that the performance of downstream tasks (including link prediction) on the original graph remains robust to perturbations
>
> >Why Th2 can be applied to higher-degree polynomial as filter/non-linearity?
>
> Thank you for this question. The statement was clarified in the revised pdf, including the precise bound that applies to any-degree SPGNNs. The crux to deal with non-linearity is that a high-degree polynomial can be unfolded as a sequence of 1-degree ones: $a_n X^n+a_{n-1} X^{n-1}+...+a_1X+a_0 = (...((a_n X+a_{n-1})X+a_{n-2})X+...+a_1)X+a_0$. So we can represent a SPGNN$\Psi$ using filters of any degree as an SPGNN$\Phi$ using 1-degree polynomials, so with a higher number of parameters. In addition to the formal definition, in Sec.A.4.1, we added Fig 4 to detail this transformation $\Psi$ to $\Phi$.
>
> >Th2: shift operators must be symmetric: is this unrealistic...edges are typically directed in relational databases, shift operators not symmetric?
>
> Please note that all db-to-graph transformations we investigate (existing or ours) yield undirected graphs, and our results hold for these constructions. For completeness, we have now included a formal statement for directed graphs (Sec A.4.2).
>
> >Why db-to-graph transformation?
>
> A significant body of literature, e.g. [3-5], demonstrates the effectiveness of graph-based models for downstream tasks on tabular data. Graphs can provide a natural representation of relational databases, preserving essential logical properties like isomorphism, query closure, foreign key constraints.
>
> >...theoretical upper-bound is up to 8 orders of magnitude larger than the empirical average distance, are bounds of any use?
>
> While the theoretical bound on stability is a worst-case one, the average case (after Corollary 2.1) exhibits significantly better stability. This aligns with previous research on stability [6], which also observed a gap theoretical bounds vs practical performance. As theoretical bounds often rely on conservative assumptions, experimental validation is crucial. Despite the bound's looseness, our theoretical work gives 2 valuable insights: (i) the bound's form identifies key stability parameters, enabling the design of more stable graph constructions, (ii) the Lipschitz-like bound - novel contribution - which addresses a fundamental, natural question in stability theory.
>
> >...other models, Graph Transformers/GAT?
>
> Although spectral GNNs have been subject to some theoretical analysis [7], a comprehensive investigation of other GNN models was beyond the scope of this work
>
> >...distinguish stability from over-smoothing...experiments with very few layers
>
> We conducted stability experiments (paragraph Oversmoothing, Appendix B) using the same graphs from the appendix, but with a 2-layer SPGNN instead of the original one. They lead to very similar results (Fig.10-11), at slightly different scales
>
> [1] PLATONOV et al A critical look at the evaluation of GNNs under heterophily: Are we really making progress?arXiv 2023
>
> [2]ROBINSON et al Relbench:A benchmark for deep learning on relational databases.arXiv2024
>
> [3]FEY et al Relational deep learning: Graph representation learning on relational databases.arXiv2023
>
> [4]CAPPUZZO et al Creating embeddings of heterogeneous relational datasets for data integration tasks.SIGMOD20
>
> [5]SCHLICHTKRULL et al Modeling relational data with graph convolutional networks.ESWC18
>
> [6]GAMA et al Stability properties of graph neural networks.Signal Processing2020
>
> [7]HUANG et al On the stability of expressive positional encodings for graph neural networks.arXiv2023

---

> ### Author Response · Authors · 2024-11-26
>
> Dear Reviewer,
>
> We hope we have adequately addressed all the points raised in your review. We remain open to further discussion on any remaining concerns before the discussion period concludes.
>
> Sincerely,
> The Authors

---

### Official Review · Reviewer_jor2 · 2024-11-03

**Soundness:** 3
**Presentation:** 2
**Contribution:** 2
**Rating:** 5
**Confidence:** 4

**Summary:**

The work poses in the topic of GNN and stability of learned embeddings. In particular the aim is to study the stability of embeddings learned by GNNs in relational databases and to present some techniques to transform a relational database into a graph in order to apply message passing schema

**Strengths:**

- The theoretical foundation supporting this work is robust, making it a crucial component.
-  Additionally, the problem being addressed is highly relevant.

**Weaknesses:**

- The paper lacks clarity in several areas, which I will specify in my subsequent questions.
- While the paper argues for treating a relational database as a homogeneous graph, it does not adequately explain why this is the case. Since this claim is significant, I recommend that the authors clarify it further.
- The experimental section is limited by the absence of a recent benchmark [1] related to relational databases, which I suggest the authors incorporate into their experiments.
- The paper does not address various models typically used for heterogeneous graphs as competitors.

**Questions:**

**Main**

- In the final part of the Introduction (**our second contribution**), you mention that current database-to-graph construction methods often result in overly heterogeneous structures. However, it’s unclear why this is problematic. Given the importance of this claim, please provide more justification.
- In the last paragraph of Related Works, you mention approaches that use LLMs to generate embeddings from databases but don’t cite them or discuss their limitations. Please elaborate on these methods and, if relevant, include a comparison or clarify why a comparison isn’t necessary.
- In the definition of SPGNN (3.2) in the node embeddings paragraph, you state that the $L-$th layer of SPGNN does not output embeddings. What does it output, then? This part is unclear, as $X^{(l)}$ was previously defined as the feature map at layer $l$.
- In Section 5, could you clarify why you describe the first two database-to-graph transformation methods as heterophilic? They seem more accurately described as heterogeneous rather than heterophilic. Additionally, you mention that these methods fail to capture the underlying structure of the database, but the reasoning behind this could be expanded right now, it’s a bit unclear.
- In Section 5, under **Perturbations**, you suggest two strategies for handling size mismatches in shift operators. In the second, you propose keeping deleted nodes but removing their connections to maintain node count—this works for tuple removal, but what about when tuples are added, as often occurs in relational databases?
- In relational databases, nodes are often both added and removed over time. In your experiments, however, you only demonstrate node removal. Could you show both additions and removals simultaneously? This would better support your model’s stability in relational database scenarios.
- There is a recent benchmark [1] that proposes a way of constructing graph from relational databases that you don’t compare with.
- Since treating a relational database as a homogeneous graph is not very convincing, it would be interesting to compare relational GNN approaches with your model when considering databases as heterogeneous.

**Minors**
- The introduction lacks citations in several parts. For instance, when you state, “Recent advances in deep learning, particularly Graph Neural Networks, have shown effectiveness in learning from tabular, relational data,” you should support this with references. The same applies to other paragraphs in this section.
- The "stability" subsection in Related Works feels disjointed, and the connection between each paragraph and the current work is unclear.
- In section 4.1 you write that as limitations of the previous work is that the result is local. It is not very clear how you define local in this context


[1] Robinson et al., Relbench: A benchmark for deep learning on relational databases, 2024

---

> ### Author Response · Authors · 2024-11-21
> **rebuttal on review jor2**
>
> > In Introduction ... you mention that current database-to-graph construction methods often result in overly heterogeneous structures. \textbf{However, it’s unclear why this is problematic}... justification.
>
> We regret any confusion caused by our previous statement. We aim to avoid or minimize heterophily in graphs derived from relational databases, which are *inherently heterogeneous* (with several types of nodes), since this characteristic can hinder the
> effectiveness of GNNs (e.g., as discussed in the three references below). In contrast, existing approaches, such as the tripartite graph construction EmBDi, typically yield heterophilic graphs, hence with both properties (heterogeneous and heterophilic) simultaneously. The revised pdf addresses this discrepancy.
>
> Jiong Zhu et al.Graph neural networks with heterophily.AAAI21.
>
> Dongxiao He et al.Block modeling-guided graph convolutional neural networks.AAAI22.
>
> Tao Wang et al.Powerful graph convolutional networks with adaptive propagation mechanism for homophily and heterophily.AAAI22.
>
> > ... you mention approaches that use LLMs to generate embeddings from databases but don’t cite them or discuss their limitations. Please elaborate...
>
> We would like to stress here the importance of GNNs and GNN embeddings for database applications, supported by the rich line of recent research that directly applies GNNs to databases for various downstream tasks. Our theoretical and experimental findings contribute to this field. While other embedding methods are available, GNNs provide a strong theoretical foundation and are significantly more efficient and economical than LLMs.
>
> > In Sec 5, clarify why you describe the first two database-to-graph transformation methods as heterophilic? They seem more accurately described as heterogeneous rather than heterophilic.
>
> We will clarify next why heterophilic is the right term here. The term **heterogeneous graphs** refers to graphs which have different types / semantics attached to their nodes. For example, in our relational-to-graph transformations, these types can represent tuples, attributes, values, relations, or tuple-value pairs.
>
> The term **heterophilic graphs** refers to heterogeneous graphs where nodes of different types are more likely to be connected. For example, in bipartite (or multi-partite) graphs constructed from relational databases using existing approaches (e.g., [8]), a node connects only to others having a different type.
>
> > Additionally, you mention that these methods fail to capture the underlying structure of the database ... expand reasoning
>
> Graphs derived from databases that do not fully capture the underlying database structure can drastically affect the performance of downstream tasks. For one example, the EmBDi (tripartite) graph cannot differentiate between two **distinct** tuples that happen to involve the same values (e.g., FirstName: John, LastName: Oliver vs. FirstName: Oliver, LastName: John).
>
> > In Sec 5, Perturbations, you suggest two strategies for handling size mismatches in shift operators. In the second, you propose keeping deleted nodes but removing their connections to maintain node count... what about when tuples are added ?
>
> Please note that the case of tuple addition is **symmetric** to the one of tuple removal: adding a tuple to a database instance $D$ to obtain an instance $D'$ is the exact opposite operation of removing the added tuple from $D'$ to obtain $D$. As required, the perturbation metric must be symmetric in the two input databases. Therefore, these two operations present no conceptual differences and without loss of generality we did not discuss further the case of tuple addition.  We have clarified this point in the revised version of the paper.
>
> > In relational databases, nodes are often both added and removed ... Could you show both additions and removals simultaneously?
>
> We added in the appendix new experiments where both tuple deletions and additions occur (Fig. 15-17). The results show that embeddings remain stable even under both kinds of modifications.
>
> > There is a recent benchmark [1] that proposes a way of constructing graphs from relational databases that you don’t compare with.
>
> [1] Robinson et al. Relbench: A benchmark for deep learning on relational databases. arXiv 2024.
>
> Thank you for pointing out this recent paper, released around the time of our submission. Indeed, all the datasets presented there can be used within our stability framework. For databases that are temporal, we could  apply our framework to specific snapshots thereof.
>
> > Since treating a relational database as a homogeneous graph is not very convincing, it would be interesting to compare relational GNN approaches with your model when considering databases as heterogeneous.
>
> We stress that the database graphs we consider **are not homogeneous**. Database graphs obtained using our transformations are all heterogeneous. This has been clarified further in the revised version of the paper.

---

> > ### Comment · Reviewer_jor2 · 2024-11-25
> >
> > 1. Heterogeneous and heterophilic are not the same. A graph can be heterogeneous yet exhibit high homophily through specific relational paths (e.g., an author A1 connected to a paper P, which is connected to author A2; if A1 and A2 have similar features/labels, the graph remains homophilic). Heterogeneous graphs provide additional information about the types of relations connecting nodes, which can be leveraged using heterogeneous GNNs.
> >
> > 2. Can you demonstrate that GNN methods are more efficient and cost-effective than LLMs in this context? A claim like this requires supporting evidence, such as a table with results. Additionally, the statement lacks citations, at least one supporting the claim.
> >
> > 3. The phrase "The term heterophilic graphs refers to heterogeneous graphs where nodes of different types are more likely to be connected" seems to imply that heterogeneous graphs are inherently heterophilic. However, heterogeneous graphs are not inherently heterophilic. Homophily is simply measured differently in such cases. Refer to [1], Section 3.2, for further explanation.
> >
> > 4.  Thanks for the clarification
> >
> > 5. My comment on node addition/removal relates to the example where, during node removal, you keep the node in the count, ensuring the matrices remain the same size (essentially retaining the node but removing all its edges). However, node addition is different, as the total number of nodes is unknown in advance, with new nodes introduced over time. I still don’t fully understand how this is handled (referring to lines [412-419] in the paper).
> >
> > 6. Thanks for the new experiments
> >
> > 7. The paper I cited was accepted around June-July, a couple of months before the deadline. Since it includes some datasets, it would have been beneficial to include a few experiments using them.
> >
> > 8. There is extensive literature on using Heterogeneous GNNs for heterogeneous graphs [2-5]. Since your final graphs are heterogeneous, I believe it’s crucial to include such methods in your experiments.
> >
> > [1] The Heterophilic Graph Learning Handbook
> >
> > [2] Ziniu Hu et al. Heterogeneous graph transformer. 2020
> >
> > [3] Qingsong Lv et al. Are we really making much progress? re we really making much progress? revisiting, benchmarking and refining heterogeneous graph neural networks. 2021
> >
> > [4] Michael Schlichtkrull et al Modeling relational data with graph convolutional networks.  2018
> >
> > [5] Le Yu et al Heterogeneous graph representation learning with relation awareness. 2022

---

> ### Author Response · Authors · 2024-11-25
> **answers to additional comments**
>
> Thank you for engaging in the discussion during this rebuttal phase and for the additional requests for clarifications.
>
> >1. Heterogeneous and heterophilic are not the same. A graph can be heterogeneous yet exhibit high homophily...
>
> It seems that we do not use the same definition for *heterogeneous graphs*. In our work, as indicated in the paper and in the rebuttal, we used the term *heterogeneous* for graphs having different types of nodes, but only one type of edges.
>
> >2. Can you demonstrate that GNN methods are more efficient and cost-effective than LLMs...
>
> This statement deserves a more detailed justification, indeed. In terms of both data size and model size, it is **generally accepted that LLMs are much more expensive to train,  maintain, or adapt to different application contexts.** **LLMs** often have billions of parameters, requiring substantial computational resources for training; they require massive datasets, often in the terabytes or even petabytes range, to learn complex language patterns. **GNNs** typically have fewer parameters, especially when compared to large LLMs, making them less computationally intensive; they  can be trained on smaller datasets, especially for domain-specific tasks, reducing data acquisition and preprocessing costs, as discussed in the recent survey below.
>
> Ren et al. A Survey of Large Language Models for Graphs.KDD24
>
> >3. The phrase "The term heterophilic graphs..." seems to imply that heterogeneous graphs are inherently heterophilic...
>
> Please note that our statement does not at all mean that all heterogeneous graphs are automatically heterophilic. The phrase "where nodes of different types are more likely to be connected" is essential to understanding our meaning: a graph needs to heterogeneous AND have a high probability of connections between nodes of different types in order to be heterophilic.
>
> > 5. ...on node addition/removal...addition is different...
>
> As previously mentioned, we do not see node addition and node deletion as distinct issues, neither in the theory nor in the implementation. Here is how we proceed for additions: let us consider a graph $\mathcal{G}$ and its shift operator $\mathbf{S}$, as well as a graph $\hat{\mathcal{G}}$  with its shift operator $\hat{\mathbf{S}}$, such that  $\hat{\mathcal{G}}$ is obtained from $\mathcal{G}$ by adding one node $n$ along with edges connecting it to other nodes. We need to identify the integer $i$ such that the $i$-th row/column of  $\hat{\mathbf{S}}$  corresponds to the edges of $n$. Then, we would add a row/column of 0s in $\mathbf{S}$ at the $i$-th position, so that it matches the shape of $\hat{\mathbf{S}}$ (this is conceptually as if $n$ was already in $\mathcal{G}$, but unconnected to the rest of the graph). We would use this new matrix $\mathbf{S}$ to compare the embeddings of nodes in both graphs. Deletion amounts to the symetrical construction.  When we consider a chain of additions/deletions, we simply iterate such constructions.
>
> While it is true that we do not know in advance the final number of nodes, following additions, we always compute the bound after all the modifications are made, at the end of the process. **This means that when we compute the bound, we indeed have access to the set of added / deleted nodes, as in the case where we only consider node deletions.** Hence there is no specific difficulty on this aspect.
>
> >7. The paper I cited was accepted around June-July, a couple of months before the deadline. Since it includes some datasets, it would have been beneficial to include a few experiments using them
>
> The paper was indeed uploaded to arXiv on July 29th;  we are currently unaware of its official acceptance date. We are of course open to incorporating additional datasets into our experimental framework, in a revised version. We do believe that the datasets we currently use are sufficiently representative to demonstrate the stability of embeddings generated through db-to-graph transformations. Our experiments encompass a variety of graph types, including citation networks and graphs derived from relational databases. These databases span diverse domains, from industry-standard benchmarks like TPC-E to specialized fields such as medicine and geography. Furthermore, our experiments cover both homogeneous (citation networks) and heterogeneous datasets.
>
> >8. There is extensive literature on using Heterogeneous GNNs for heterogeneous graphs...include such methods in your experiments
>
> As mentioned in our answer to (1), it seems that we do not use the same definition for heterogeneous graphs.  The methods you cite focus on heterogeneous graphs with directed **labeled** edges. Our work applies to graphs where the edges are non labeled, which are still heterogeneous according to the definition used in our paper. The extension of our work to more specific classes of GNNs, including Heterogeneous GNNs, is indeed very interesting for future work, but it is out of the scope of this paper.

---

> > ### Comment · Reviewer_jor2 · 2024-12-02
> >
> > I thank the authors for their clarifications.
> > However, I remain skeptical about the claim that no heterogeneous GNN methods are applied in this work. Specifically, when a graph contains different types of nodes, the edge types can often be inferred from the types of the connected nodes. For instance, in a graph with node types such as author (A), paper (P), and conference (C), the edge types could naturally be defined based on the types of the adjacent nodes.
> > To ensure the work is comprehensive, I would expect a comparison with such approaches as well.

---

> ### Author Response · Authors · 2024-12-02
>
> We wish to emphasize that the proposed graph databases are designed to precisely replicate the structure and logical relationships found in the original relational databases they represent. This structural and logical similarity is a **key factor** in generating meaningful embeddings.
>
> Unfortunately, HGCNs often perform poorly in capturing the overall graph structure when meta-paths (paths capturing interactions between nodes of different types via directed edges) are automatically determined [4-6] (i.e., not pre-specified by users [1-3]). To avoid this **known and well-documented limitation** of HGCNs, we prioritize in this work classical GNNs, which are **more effective** in extracting both local and global information from the graph while maintaining **computational efficiency** [7].
>
> Please note as well that a substantial body of literature exists on GNNs for graphs lacking edge labels -- especially those that are not RDF-like or RDF-compatible -- and is continuously enriched by new studies. This literature remains relevant and is in no way superseded by research on heterogeneous GNNs.
>
> Furthermore, we wish to bring to your kind attention a key aspect of our work: the concept of stability is fundamental to the robust performance of ANY neural model including GNNs, regardless of the type, viz., homogenous or heterogeneous (whether heterophilic or not). One can certainly make comparative evaluations with all types of GNNs, but these comparisons are necessarily empirical. The importance of stability is theoretical and fundamental and it plays a central role in our work. Unfortunately this does not seem to be recognized either in the review or in the rebuttal discussion. In our paper, we have chosen to use GNNs applied to graphs obtained from transformations to relational databases to showcase the importance of stability. We ask that you kindly take this into account.
>
> [1] WANG, Xiao, JI, Houye, SHI, Chuan, et al. Heterogeneous graph attention network. WWW 2019
>
> [2] ZHOU, Sheng, BU, Jiajun, WANG, Xin, et al. HAHE: Hierarchical attentive heterogeneous information network embedding. arXiv preprint arXiv:1902.01475, 2019.
>
> [3] WANG, Shen, CHEN, Zhengzhang, LI, Ding, et al. Attentional heterogeneous graph neural network: Application to program reidentification. In SIAM SDM 2019
>
> [4] ZHANG, Yizhou, XIONG, Yun, KONG, Xiangnan, et al. Deep collective classification in heterogeneous information networks. In WWW 2018
>
> [5] CHEN, Xia, YU, Guoxian, WANG, Jun, et al. ActiveHNE: Active heterogeneous network embedding. arXiv preprint arXiv:1905.05659, 2019.
>
> [6] ZHANG, Chuxu, SONG, Dongjin, HUANG, Chao, et al. Heterogeneous graph neural network. In KDD 2019
>
> [7] YANG, Yaming, GUAN, Ziyu, LI, Jianxin, et al. Interpretable and efficient heterogeneous graph convolutional network. IEEE TKDE 2021

---

### Official Review · Reviewer_6hKW · 2024-11-04

**Soundness:** 3
**Presentation:** 3
**Contribution:** 3
**Rating:** 6
**Confidence:** 2

**Summary:**

This paper theoretically examines the stability of GNNs in the context of significant graph perturbations, a common challenge in relational databases. By comparing existing database-to-graph construction methods, the paper proposes a new method designed for improved stability, demonstrating practical advantages.

**Strengths:**

1. This paper expands the applicability of current stability bounds to a broader range of settings, making it more relevant for practical relational database use.
2. It introduces a new method for converting relational databases to graphs with enhanced stability.
3. The experiments are conducted on various datasets, supplemented by empirical analyses to validate GNN effectiveness.

**Weaknesses:**

1. The selected graph datasets are limited to a single domain (citation networks) and type (highly homogeneous graphs). This may restrict the generalizability of the method to other scenarios, such as heterogeneous graphs datasets from HGB [1]

[1] Are we really making much progress? Revisiting, benchmarking, and refining heterogeneous graph neural networks https://arxiv.org/abs/2112.14936

**Questions:**

1. Many large relational datasets are temporal, so predictions must follow causal constraints. This means future events cannot be used when making predictions. To handle this, temporal sampling is often necessary within GNNs. It is important to consider whether this method can work well in a temporal setting. Specifically, when predicting for a node, different nodes and time embeddings may be included in the sampled subgraph due to varying time constraints. How would these modifications affect stability?

---

> ### Author Response · Authors · 2024-11-21
> **rebuttal on review 6hKW**
>
> > The selected graph datasets are limited to a single domain (citation networks) and type (highly homogeneous graphs). This may restrict the generalizability of the method to other scenarios, such as heterogeneous graph datasets from HGB [1].
>
> Please note that while our first experimental stage uses citation network graphs, our second (and primary) experimental investigation involves graphs obtained from relational databases. These databases cover a wide range, from the industry-standard TPC-E benchmark to specialized domains such as medical and geographical data. Therefore, **our experiments do cover both homogeneous (citation networks) and heterogeneous datasets**. For the latter, we recall the type of graphs that are obtained from a relational database through graph transformations. For instance, the 5-DB relational-to-graph transformation produces a heterogeneous graph with 5 different node types (Relation names, AT, Tuple, Attribute names, Values)  and 4 edge types  (R-T, T-AT, AT-A, AT-V).
>
> > Many large relational datasets are temporal, so predictions must follow causal constraints. This means future events cannot be used when making predictions. To handle this, temporal sampling is often necessary within GNNs. It is important to consider whether this method can work well in a temporal setting. Specifically, when predicting for a node, different nodes and time embeddings may be included in the sampled subgraph due to varying time constraints. How would these modifications affect stability?
>
> Thank you for this question. First, please note that stability is a desirable property of embeddings, regardless of whether they are obtained from static data or temporal, evolving data. Most databases today remain relational and primarily focus on the current state (aka snapshot) of the data, rather than its historical evolution or future state predictions. While interest in temporal and time-series databases is growing, traditional relational databases still dominate the market by far. While we haven't explored the temporal aspect of relational databases in this work, we believe that understanding the resilience of embeddings to incomplete data is crucial. Tasks like missing value imputation and tuple classification rely heavily on stable embeddings. As missing data increases, unstable embeddings can significantly impact downstream accuracy (see [1]).
>
> Investigating the stability of embeddings in a temporal database is certainly a promising avenue for future research.
>
> [1] SCHUMACHER et al. The effects of randomness on the stability of node embeddings. ECML/PKDD 2021

---

> > ### Comment · Reviewer_6hKW · 2024-11-26
> >
> > Thanks for your response. I will maintain my positive rating.

---

### Author Response · Authors · 2024-11-21
**general comment to all reviewers**

We thank all the reviewers for their extensive comments and questions.

Following the reviewers' suggestions, we have included additional experiments on (i) sequences of tuple additions / deletions in a relational database and their effects on stability on the GNN embeddings of the database-to-graph constructions (Fig.  15-17) and  (ii) the potential effect of over-smoothing in the GNNs (Fig. 10-11 for the comparison on the Mutagenesis dataset).  These experiments further consolidate the main observations of our initial empirical evaluation.

The edits for all minor comments / clarifications were done directly in the revised  paper. Next, we provide our answers to the detailed questions.

---

### Meta-Review · Area_Chair_wwBp · 2024-12-21

**Metareview:**

This manuscript analyzes the stability of signal-processing GNNs (SPGNNs) by establishing limits for both node-level and graph-level embeddings. The stability is related to how much the embeddings generated by a GNN change when the input graph undergoes modifications. This manuscript presents some techniques to transform a relational database into a graph to apply message-passing schema.

Reviewers commented that the topic is interesting and that the problem being addressed is highly relevant to the venue. However, some critical issues make this manuscript incomplete, and the manuscript needs to be reviewed again after major revisions. The authors should consider the following points in the revisions.
- Compare with Heterogeneous GNNs in experiments
- Compare with a recent benchmark [1] that proposes a way of constructing graphs from relational databases
- Clarify terminologies and details questioned by the reviewers.
- The paper's topic and focus are two-folded and loosely related. Relate the two seemingly separable stories and make the manuscript more coherent.

[1] Robinson et al. Relbench: A benchmark for deep learning on relational databases. arXiv 2024.

**Additional Comments On Reviewer Discussion:**

Reviewer 6hKW's review and comments were down-weighted in making the decision since this reviewer explicitly stated the unfamiliarity with the topic and indicated a low confidence score. Reviewer jor2 and Reviewer V2r2 left detailed reviews and questions. Also, they pointed out some critical points that the authors should address when revising the manuscript.

---

### Decision · Program_Chairs · 2025-01-22

Reject